# Private (Stochastic) Non-Convex Optimization Revisited: Second-Order Stationary Points and Excess Risks

**Arun Ganesh**
Google Research
arunganesh@google.com

**Daogao Liu**
University of Washington[*]
dgliu@uw.edu

**Sewoong Oh**
University of Washington and Google Research
sewoong@cs.washington.edu

**Abhradeep Thakurta**
Google DeepMind
athakurta@google.com

## Abstract

We reconsider the challenge of non-convex optimization under differential privacy constraint. Building upon the previous variance-reduced algorithm SpiderBoost, we propose a novel framework that employs two types of gradient oracles: one that estimates the gradient at a single point and a more cost-effective option that calculates the gradient difference between two points. Our framework can ensure continuous accuracy of gradient estimations and subsequently enhances the rates of identifying second-order stationary points. Additionally, we consider a more challenging task by attempting to locate the global minima of a non-convex objective via the exponential mechanism without almost any assumptions. Our preliminary results suggest that the regularized exponential mechanism can effectively emulate previous empirical and population risk bounds, negating the need for smoothness assumptions for algorithms with polynomial running time. Furthermore, with running time factors excluded, the exponential mechanism demonstrates promising population risk bound performance, and we provide a nearly matching lower bound.

## 1 Introduction

Differential privacy [19] is a standard privacy guarantee for training machine learning models. Given a randomized algorithm $\mathcal{A} : P^* \to R$, where $P$ is a data domain and $R$ is a range of outputs, we say $\mathcal{A}$ is $(\varepsilon, \delta)$-differentially private (DP) for some $\varepsilon \geq 0$ and $\delta \in [0, 1]$ if for any neighboring datasets $\mathcal{D}, \mathcal{D}' \in P^*$ that differ in at most one element and any $\mathcal{R} \subseteq R$, the distribution of the outcome of the algorithm, e.g., pair of models trained on the respective datasets, are similar:

$$\Pr_{x \sim \mathcal{A}(\mathcal{D})}[x \in \mathcal{R}] \leq e^\varepsilon \Pr_{x \sim \mathcal{A}(\mathcal{D}')}[x \in \mathcal{R}] + \delta.$$

Smaller $\varepsilon$ and $\delta$ imply the distributions are closer; hence, an adversary accessing the trained model cannot tell with high confidence whether an example $x$ was in the training dateset. Given this measure of privacy, we consider the problem of optimizing a non-convex loss while ensuring a desired level of privacy. In particular, suppose we are given a dataset $\mathcal{D} = \{z_1, \ldots, z_n\}$ drawn i.i.d. from underlying distribution $\mathcal{P}$. Each loss function $f(\cdot; z) : \mathcal{K} \to \mathbb{R}$ is $G$-Lipschitz over the convex set $\mathcal{K} \subset \mathbb{R}^d$ of diameter $D$. Let the population risk function be $F_{\mathcal{P}}(x) := \mathbb{E}_{z \sim \mathcal{P}}[f(x; z)]$ and the empirical risk function be $F_{\mathcal{D}}(x) := \frac{1}{n} \sum_{z \in \mathcal{D}} f(x; z)$. We also denote $F_S(x) := \frac{1}{|S|} \sum_{z \in S} f(x; z)$ for $S \subseteq \mathcal{D}$.

---

[*]Most of this work was done while the author was an intern at Google.

37th Conference on Neural Information Processing Systems (NeurIPS 2023).

Our focus is in minimizing non-convex (empirical and population) risk functions, which may have multiple local minima. Since finding the global optimum of a non-convex function can be challenging, an alternative goal in the field is to find stationary points: A first-order stationary point is a point with a small gradient of the function, and a second-order stationary point is a first-order stationary point where additionally the function has a positive or nearly positive semi-definite Hessian. As first order stationary points can be saddle points or even a local maximum, we focus on the problem of finding a second order stationary point, i.e., a local minimum, privately. Existing works in finding approximate SOSP privately only give guarantees for the empirical function $F_\mathcal{D}$. We improve upon the state-of-the-art result for empirical risk minimization and give the first guarantee for the population function $F_\mathcal{P}$. This requires standard assumptions on bounded Lipschitzness, smoothness, and Hessian Lipschitzness, which we make precise in Section 2 and in Assumption 3.1.

Compared to finding a local minimum, finding a global minimum can be extremely challenging. We also present two methods, polynomial and exponential time, that outperform existing guarantees measured in excess risks for respective computational complexities. Our primary results are succinctly summarized in Table 1.

**Related Work.** We propose a novel and simple framework based on SpiderBoost [52], and its private version [2] that achieves the current best rate for finding the first order stationary point privately. We discuss the primary difference between our framework and theirs, that is their algorithms only promise small gradient estimation errors on average, but our framework can ensure small estimation errors consistently throughout all the iterations, and the motivation behind this briefly.

In SGD and its variants, the typical approach involves obtaining an estimation $\Delta_t$ of the gradient $\nabla f(x_t)$. In the stochastic variance-reduced algorithm SpiderBoost [52, 2], it queries the gradient $\mathcal{O}_1(x_t) \approx \nabla f(x_t)$ directly every $q$ steps with some oracle $\mathcal{O}_1$, and for the other $q-1$ steps within each period, it queries the gradient difference between two steps, that is $\mathcal{O}_2(x_t, x_{t-1}) \approx \nabla f(x_t) - \nabla f(x_{t-1})$, and maintain $\Delta_t = \Delta_{t-1} + \mathcal{O}_2(x_t, x_{t-1})$. The contrast between these two types of oracles can be perceived as $\mathcal{O}_1$ being more accurate but also more costly, in terms of computation or privacy budget, although our framework does not strictly necessitate this assumption.

As SpiderBoost queries $\mathcal{O}_1$ every $q$ steps, the error on the estimation may accumulate and $\|\Delta_t - \nabla f(x_t)\|$ can become large. Despite this, as demonstrated in [2], these estimations can, on average, suffice to find a private FOSP. However, such large deviations pose a challenge when scrutinizing behavior near a saddle point. For instance, when the current point is a saddle point, but the current estimation is unsatisfactory, it becomes uncertain whether the algorithm can escape the saddle point. It could be argued that average good estimations could achieve a SOSP, but to the best of our knowledge, there is no existing result addressing this concern.

A plausible solution to this challenge is to maintain high-quality gradient estimations throughout all iterations, a feat accomplished by our framework. We believe this feature holds promise for improving the outcomes of various other optimization problems, thus enhancing the overall appeal and significance of our work.

## 1.1 Main Results

**SOSP.** One of our main contributions is a refined optimization framework (Algorithm 1), predicated on the variance-reduced SpiderBoost [52], which guarantees consistently accurate gradient estimations. By integrating this framework with private gradient oracles, we achieve improved error rates for privately identifying SOSP of both empirical and population risks.

Advances in private non-convex optimization have focused on finding a first-order stationary point (FOSP), whose performance is measured in ($i$) the norm of the empirical gradient at the solution $x$, i.e., $\|\nabla F_\mathcal{D}(x)\|$, and ($ii$) the norm of the population gradient, i.e., $\|\nabla F_\mathcal{P}(x)\|$. We survey the recent progress in the appendix in detail.

**Definition 1.1** (First-order stationary point). *We say $x \in \mathbb{R}^d$ is a First-Order Stationary Point (FOSP) of $g : \mathbb{R}^d \to \mathbb{R}$ iff $\nabla g(x) = 0$. $x$ is an $\alpha$-FOSP of $g$, if $\|\nabla g(x)\|_2 \leq \alpha$.*

Since FOSP can be a saddle point or a local maxima, finding a second-order stationary point is desired. Exact second-order stationary points can be extremely challenging to find [25]. Instead, progress is commonly measured in terms of how well the solution approximates an SOSP.

| | $\alpha$-SOSP | | Excess population risk | |
|---|---|---|---|---|
| | empirical | population | poly-time | exp-time |
| SOTA | $\min(\frac{d^{\frac{1}{4}}}{n^{\frac{1}{2}}\varepsilon^{\frac{1}{2}}}, \frac{d^{\frac{4}{7}}}{n^{\frac{4}{7}}\varepsilon^{\frac{4}{7}}})$ | N/A | $\frac{d}{\varepsilon^2 \log n}$ ♠ | N/A |
| Ours | $\frac{d^{\frac{1}{3}}}{n^{\frac{2}{3}}\varepsilon^{\frac{2}{3}}}$ | $\frac{1}{n^{\frac{1}{3}}} + \left(\frac{\sqrt{d}}{n\varepsilon}\right)^{\frac{3}{7}}$ | $\frac{d \log\log n}{\varepsilon \log(n)}$ | $\frac{d}{n\varepsilon} + \sqrt{\frac{d}{n}}$ |
| LB | $\frac{\sqrt{d}}{n\varepsilon}$ | $\frac{1}{\sqrt{n}} + \frac{\sqrt{d}}{n\varepsilon}$ | $\frac{d}{n\varepsilon} + \sqrt{\frac{d}{n}}$ | $\frac{d}{n\varepsilon} + \sqrt{\frac{d}{n}}$ |

Table 1: SOTA refers to the best previously known bounds on $\alpha$ for $\alpha$-SOSP by [46, 48] and on the excess population risk by [46]. We introduce algorithm 1 that finds an $\alpha$-SOSP (columns 2–3) with an improved rate. We show exponential mechanism can minimize the excess risk in polynomial time and exponential time, respectively (columns 4 and 5). ♠ requires extra assumption on bounded smoothness. The lower bounds for SOSP are from [2], and the lower bound on excess population risk is from Theorem 5.11. We omit logarithmic factors in $n$ and $d$ except the upper bounds for excess population risk with polynomial time.

**Definition 1.2** (Second-order stationary point, [1])**.** *We say a point $x \in \mathbb{R}^d$ is a Second-Order Stationary Point (SOSP) of a twice differentiable function $g : \mathbb{R}^d \to \mathbb{R}$ iff $\|\nabla g(x)\|_2 = 0$ and $\nabla^2 g(x) \succeq 0$. We say $x \in \mathbb{R}^d$ is an $\alpha$-SOSP for $\rho$-Hessian Lipschitz function $g$, if $\|\nabla g(x)\|_2 \leq \alpha \bigwedge \nabla^2 g(x) \succeq -\sqrt{\rho\alpha}I$ .*

On the empirical risk $F_{\mathcal{D}}$, the SOTA on privately finding $\alpha$-SOSP is by [46, 48], which achieves $\alpha = \tilde{O}(\min\{(\sqrt{d}/n)^{1/2}, (d/n)^{4/7}\})$. In Theorem 4.2, we show that applying the proposed Algorithm 1 achieves a rate bounded by $\alpha = \tilde{O}((\sqrt{d}/n)^{2/3})$, which improves over the SOTA in all regime.[2] There remains a factor $(\sqrt{d}/n)^{-1/6}$ gap to a known lower bound of $\alpha = \Omega(\sqrt{d}/n)$ that holds even if finding only an $\alpha$-FOSP [2]. On the population risk $F_{\mathcal{P}}$, applying Algorithm 1 with appropriate private gradient oracles is the first private algorithm to guarantee finding an $\alpha$-SOSP with $\alpha = \tilde{O}(n^{-1/3} + (\sqrt{d}/n)^{3/7})$ in Theorem 4.6. There is a gap to a known lower bound of $\alpha = \Omega(1/\sqrt{n} + \sqrt{d}/n\varepsilon)$ that holds even if finding only an $\alpha$-FOSP [2].

**Minimizing Excess Risk.** In addition to the optimization framework, we present sampling-based algorithms designed to identify a private solution $x^{priv} \in \mathbb{R}^d$ that minimizes both the excess empirical risk: $\mathbb{E}[F_{\mathcal{D}}(x^{priv})] - \min_{x \in \mathcal{K}} F_{\mathcal{D}}(x)$, and the excess population risk: $\mathbb{E}[F_{\mathcal{P}}(x^{priv})] - \min_{x \in \mathcal{K}} F_{\mathcal{P}}(x)$. Here, the expectation is over the randomness of the solution $x^{priv}$ and the drawing of the training date over $\mathcal{P}$. Our method is different from [46], which Gradient Langevin Dynamics and achieves in polynomial time a bound of $O(d\sqrt{\log(1/\delta)}/(\varepsilon^2 \log n))$ for both excess empirical and population risks with a need for the smoothness assumption. In Table 1 we omit excess empirical risk, as the bounds align with those of the population risk. We introduce a sampling-based algorithm from the exponential mechanism, which runs in polynomial time and achieves excess empirical and population risks bounded by $O(d\sqrt{\log(1/\delta)}/(\varepsilon \log(nd)))$ with improved dependence on $\varepsilon$ (Theorem 5.6). Crucially, it achieves these results without the need for the smoothness assumption required by [46].

In the case of permitting an exponential running time, [23] demonstrated $\tilde{O}(d/(\varepsilon n))$ upper bound for non-convex excess empirical risks alongside a nearly matching lower bound. However, establishing a tight bound for the excess population risk remained an unresolved problem. We address this open question by providing nearly matching upper and lower bounds of $\tilde{\Theta}(d/(\varepsilon n) + \sqrt{d/n})$ for the excess population risk (Theorem 5.8).

## 1.2 Our Techniques

**Stationary Points.** In our framework, we deviate from the traditional approach of querying $\mathcal{O}_1$ once every $q$ steps. Instead, we introduce a novel but simple method of monitoring the total drift we make, that is $\text{drift}_t = \sum_{i=\tau_t}^{t} \|x_i - x_{i-1}\|_2^2$, where $\tau_t$ represents the last timestamp when we employed $\mathcal{O}_1$. As we are considering smooth functions, the maximum error to estimate $\nabla f(x_t) - \nabla f(x_{t-1})$

---

[2]We want $\alpha = o(1)$ and hence can assume $d \leq n^2$.

is proportional to $\|x_t - x_{t-1}\|_2$. If the value $\mathrm{drift}_t$ is small, we know the current estimation should still be good enough, eliminating the need for an expensive fresh estimation from $\mathcal{O}_1$. Conversely, when $\mathrm{drift}_t$ is large, the gradient estimation error may be substantial, necessitating a query to $\mathcal{O}_1$ and thus obtaining $\Delta_t = \mathcal{O}_1(x_t)$. To effectively manage the total cost, it is crucial to set an appropriate threshold to decide when the drift is significant. A smaller threshold would ensure more accurate estimations but might incur higher costs due to more frequent queries to $\mathcal{O}_1$.

Our aim is to bound the total occurrences of the event that $\mathrm{drift}_t$ is large, which leads to querying $\mathcal{O}_1$. A crucial observation is that, if $\mathrm{drift}_t$ increases rapidly, then the gradient norms are large and hence function values decrease quickly, which we know does not happen frequently under the standard assumption that the function is bounded.

In our framework, we assume $\mathcal{O}_1(x)$ is an unbiased estimation of $\nabla f(x)$, and $\mathcal{O}_1(x) - \nabla f(x)$ is Norm-SubGaussian (Definition 2.2), and similarly $\mathcal{O}_2(x, y)$ is an unbiased estimation of $\nabla f(x) - \nabla f(y)$ whose error is also Norm-SubGaussian. In the empirical case, we can simply add Gaussian noises with appropriately chosen variances to the gradients of the empirical function $\nabla F_{\mathcal{D}}$ for simplicity, and one can choose a smaller batch size to reduce the computational complexity. In the population case, we draw samples from the dataset without replacement to avoid dependence issues, and add the Gaussian noises to the sampled gradients. Hence we only need the gradient oracle complexity to be linear in the size of dataset for the population case.

**Minimizing Excess Risk.**  Our polynomial time approach harnesses the power of the Log-Sobolev Inequality (LSI) and the classic Stroock perturbation lemma. The previous work of [39] shows that if the density $\exp\left(-\beta F_{\mathcal{D}}(x) - r(x)\right)$ satisfies the LSI for some regularizer $r$, then sampling a model $x$ from this density is DP with an appropriate $(\varepsilon, \delta)$. If $r$ is a $\mu$ strongly convex function, then the density proportional to $\exp(-r)$ satisfies LSI with constant $1/\mu$, and $\exp(-\beta F_{\mathcal{D}}(x) - r(x))$ satisfies LSI with constant $\exp(\max_{x,y} |F_{\mathcal{D}}(x) - F_{\mathcal{D}}(y)|)/\mu$ by the Stroock perturbation lemma. Our bound on the empirical risk follows from choosing the appropriate inverse temperature $\beta$ and regularizer $r$ to satisfy $(\varepsilon, \delta)$-DP. The final bound on the population risk also follows from LSI, which bounds the stability of the sample drawn from the respective distribution.

When running time is not a priority, we employ an exponential mechanism over a discretization of $\mathcal{K}$ to establish the upper bound. The empirical risk bound derives from [9], and we leverage the concentration of sums of bounded random variables to bound the maximum difference over the discretizations between the empirical and population risk. We show this is nearly tight by reductions from selection to non-convex Lipschitz optimization of [23].

### 1.3   Organization

In Section 2, we present necessary definitions and backgrounds for our work. In Section 3, we construct the optimization framework, with guarantees on finding the SOSP with two different kinds of SubGaussian gradient oracles. It's crucial to note that this framework focuses solely on optimization and does not pertain to privacy. Section 4 explores the pursuits of finding the SOSP privately by constructing private SubGaussian gradient oracles and seamlessly integrating them into the existing framework. We bound the private excess bounds in Section 5. For other preliminaries, all omitted proofs and some further discussions on related work can be found in the Appendix.

## 2   Preliminaries

Throughout the paper, if not stated explicitly, the norm $\|\cdot\|$ means the $\ell_2$ norm.

**Definition 2.1** (Lipschitz, Smoothness and Hessian Lipschitz). *Given a function $f : \mathcal{K} \to \mathbb{R}$, we say $f$ is $G$-Lipschitz, if for all $x_1, x_2 \in \mathcal{K}$, $|f(x_1) - f(x_2)| \leq G\|x_1 - x_2\|$, we say a function $f$ is $M$-smooth, if for all $x_1, x_2 \in \mathcal{K}$, $\|\nabla f(x_1) - \nabla f(x_2)\| \leq M\|x_1 - x_2\|$. and we say the function $f$ is $\rho$-Hessian Lipschitz, if for all $x_1, x_2 \in \mathcal{K}$, we have $\|\nabla^2 f(x_1) - \nabla^2 f(x_2)\| \leq \rho\|x_1 - x_2\|$.*

**Definition 2.2** (SubGaussian, and Norm-SubGaussian). *A random vector $x \in \mathbb{R}^d$ is SubGaussian ($\mathrm{SG}(\zeta)$) if there exists a positive constant $\zeta$ such that $\mathbb{E}\, e^{\langle v, x - \mathbb{E}\, x \rangle} \leq e^{\|v\|^2 \zeta^2 / 2}$, $\forall v \in \mathbb{R}^d$. $x \in \mathbb{R}^d$ is norm-SubGaussian ($\mathrm{nSG}(\zeta)$) if there exists $\zeta$ such that $\Pr[\|x - \mathbb{E}\, x\| \geq t] \leq 2e^{-\frac{t^2}{2\zeta^2}}, \forall t \in \mathbb{R}$.*

**Fact 2.3.** *For a Gaussian $\theta \sim \mathcal{N}(0, \sigma^2 I_d)$, $\theta$ is $\mathrm{SG}(\sigma)$ and $\mathrm{nSG}(\sigma\sqrt{d})$.*

**Lemma 2.4** (Hoeffding type inequality for norm-subGaussian, [30]). *Let $x_1, \cdots, x_k \in \mathbb{R}^d$ be random vectors, and for each $i \in [k]$, $x_i \mid \mathcal{F}_{i-1}$ is zero-mean nSG($\zeta_i$) where $\mathcal{F}_i$ is the corresponding filtration. Then there exists an absolute constant $c$ such that for any $\delta > 0$, with probability at least $1 - \omega$, $\|\sum_{i=1}^k x_i\| \leq c \cdot \sqrt{\sum_{i=1}^k \zeta_i^2 \log(2d/\omega)}$, which means $\sum_{i=1}^k x_i$ is nSG($\sqrt{c \log(d) \sum_{i=1}^k \zeta_i^2}$).*

# 3 Convergence to Stationary Points: Framework

We present the optimization framework for finding SOSP in this section. It's important to emphasize that this framework is dedicated exclusively to optimization concerns, with privacy considerations being outside of its purview. The results about SOSP throughout the paper follows the assumptions of [46].

**Assumption 3.1.** *Any function drawn from $\mathcal{P}$ is $G$-Lipschitz, $\rho$-Hessian Lipschitz, and $M$-smooth, almost surely, and the risk is upper bounded by $B$.*

As discussed before, we define two different kinds of gradient oracles, one for estimating the gradient at one point and the other for estimating the gradient difference at two points.

**Definition 3.2** (SubGaussian gradient oracles). *For a $G$-Lipschitz and $M$-smooth function $F$:*
*(1) We say $\mathcal{O}_1$ is a first kind of $\zeta_1$ norm-subGaussian Gradient oracle if given $x \in \mathbb{R}^d$, $\mathcal{O}(x)$ satisfies $\mathbb{E}\,\mathcal{O}_1(x) = \nabla F(x)$ and $\mathcal{O}_1(x) - \nabla F(x)$ is nSG($\zeta_1$).*
*(2) We say $\mathcal{O}_2$ is a second kind of $\zeta_2$ norm-subGaussian stochastic Gradient oracle if given $x, y \in \mathbb{R}^d$, $\mathcal{O}_2(x, y)$ satisfies that $\mathbb{E}\,\mathcal{O}_2(x, y) = \nabla F(x) - \nabla F(y)$ and $\mathcal{O}_2(x, y) - (\nabla F(x) - \nabla F(y))$ is nSG($\zeta_2 \|x - y\|$).*

Note that we should assume $M \geq \sqrt{\rho \alpha}$ to make finding a second-order stationary point strictly more challenging than finding a first-order stationary point. We use smin($\cdot$) to denote the smallest eigenvalue of a matrix.

---

**Algorithm 1** Stochastic Spider

1: **Input:** Objective function $F$, Gradient Oracle $\mathcal{O}_1, \mathcal{O}_2$ with SubGaussian parameters $\zeta_1$ and $\zeta_2$, parameters of objective function $B, M, G, \rho$, parameter $\kappa$, failure probability $\omega$
2: Set $\gamma = \sqrt{4C(\zeta_2^2 \kappa + 4\zeta_1^2) \cdot \log(BMd/\rho\omega)}, \Gamma = \frac{M \log(\frac{dMB}{\rho\gamma\omega})}{\sqrt{\rho\gamma}}$
3: Set $\eta = 1/M, t = 0, T = BM \log^4(\frac{dMB}{\rho\gamma\omega})/\gamma^2$
4: Set drift$_0 = \kappa$, frozen $= 1, \nabla_{-1} = 0$
5: **while** $t \leq T$ **do**
6:     **if** $\|\nabla_{t-1}\| \leq \gamma \log^3(BMd/\rho\omega) \bigwedge$ frozen$_{t-1} \leq 0$ **then**
7:         frozen$_t = \Gamma$, drift$_t = 0$
8:         $\nabla_t = \mathcal{O}_1(x_t) + g_t$, where $g_t \sim \mathcal{N}(0, \frac{\zeta_1^2}{d} I_d)$
9:     **else if** drift$_{t-1} \geq \kappa$ **then**
10:        $\nabla_t = \mathcal{O}_1(x_t)$, drift$_t = 0$, frozen$_t =$ frozen$_{t-1} - 1$
11:     **else**
12:        $\Delta_t = \mathcal{O}_2(x_t, x_{t-1}), \nabla_t = \nabla_{t-1} + \Delta_t$, frozen$_t =$ frozen$_{t-1} - 1$
13:     **end if**
14:     $x_{t+1} = x_t - \eta \nabla_t$, drift$_{t+1} =$ drift$_t + \eta^2 \|\nabla_t\|_2^2, t = t + 1$
15: **end while**
16: **Return:** $\{x_1, \cdots, x_T\}$

---

Inspired by [2] who adapted the SpiderBoost algorithm for finding private FOSPs, we give a framework based on the SpiderBoost in Algorithm 1. Our analysis of Algorithm 1 hinges on three key properties we establish in this section: $(i)$ $\nabla_t$ remains consistently close to the true gradient $\nabla F(x_t)$ with high probability; $(ii)$ the algorithm is capable of escaping the saddle point with high probability, and $(iii)$ a large drift implies significant decrease in the function value, which enables us to limit the number of queries to the more accurate but costlier first kind of gradient oracle $\mathcal{O}_1$.

**Lemma 3.3.** *For any $0 \leq t \leq T$ and letting $\tau_t \leq t$ be the largest integer such that $\mathrm{drift}_{\tau_t}$ is set to be 0, with probability at least $1 - \omega/T$, for some universal constant $C > 0$, we have*

$$\|\nabla_t - \nabla F(x_t)\|^2 \leq \left(\zeta_2^2 \cdot \sum_{i=\tau_t+1}^{t} \|x_i - x_{i-1}\|^2 + 4\zeta_1^2\right) \cdot C \cdot \log(Td/\omega). \tag{1}$$

*Hence with probability at least $1 - \omega$, we know for each $t \leq T$, $\|\nabla_t - \nabla F(x_t)\|^2 \leq \gamma^2/16$, where $\gamma^2 := 16C(\zeta_2^2\kappa + 4\zeta_1^2) \cdot \log(Td/\omega)$ and $\kappa$ is a parameter we can choose in the algorithm.*

As shown in Lemma 3.3, the error on the gradient estimation for each step is bounded with high probability. Then by adding the Gaussian noise in Line 8, we can show the algorithm can escape the saddle point efficiently based on previous results.

**Lemma 3.4** (Essentially from [46])**.** *Under Assumption 3.1, run SGD iterations $x_{t+1} = x_t - \eta\nabla_t$, with step size $\eta = 1/M$. Suppose $x_0$ is a saddle point satisfying $\|\nabla F(x_0)\| \leq \alpha$ and $\mathrm{smin}(\nabla^2 F(x_0)) \leq -\sqrt{\rho\alpha}$, $\alpha = \gamma \log^3(dBM/\rho\omega)$. If $\nabla_0 = \nabla F(x_0) + \zeta_1 + \zeta_2$ where $\|\zeta_1\| \leq \gamma$, $\zeta_2 \sim \mathcal{N}(0, \frac{\gamma^2}{d\log(d/\omega)}I_d)$, and $\|\nabla_t - \nabla F(x_t)\| \leq \gamma$ for all $t \in [\Gamma]$, with probability at least $1 - \omega \cdot \log(1/\omega)$, one has $F(x_\Gamma) - F(x_0) \leq -\Omega\left(\frac{\gamma^{3/2}}{\sqrt{\rho}\log^3(\frac{dMB}{\rho\gamma\omega})}\right)$, where $\Gamma = \frac{M\log(\frac{dMB}{\rho\gamma\omega})}{\sqrt{\rho\gamma}}$.*

We discuss this lemma in the Appendix in more details. The next lemma is standard, showing how large the function values can decrease in each step.

**Lemma 3.5.** *By setting $\eta = 1/M$, we have $F(x_{t+1}) \leq F(x_t) + \eta\|\nabla_t\| \cdot \|\nabla F(x_t) - \nabla_t\| - \frac{\eta}{2}\|\nabla_t\|^2$. Moreover, with probability at least $1 - \omega$, for each $t \leq T$ such that $\|\nabla F(x_t)\| \geq \gamma$, we have*

$$F(x_{t+1}) - F(x_t) \leq -\eta\|\nabla_t\|^2/6 \leq -\eta\gamma^2/6.$$

With the algorithm designed to control the $\mathrm{drift}$ term, the guarantee for Stochastic Spider to find the second order stationary point is stated below:

**Lemma 3.6.** *Suppose $\mathcal{O}_1$ and $\mathcal{O}_2$ are $\zeta_1$ and $\zeta_2$ norm-subGaussian respectively. If one sets $\gamma = O(1)\sqrt{(\zeta_2^2\kappa + 4\zeta_1^2) \cdot \log(Td/\omega)}$, with probability at least $1 - \omega$, at least one point in the output set $\{x_1, \cdots, x_T\}$ of Algorithm 1 is $\alpha$-SOSP, where*

$$\alpha = \gamma\log^3(BMd/\rho\omega\gamma) = \sqrt{(\zeta_2^2\kappa + 4\zeta_1^2) \cdot \log(\frac{d/\omega}{\zeta_2^2\kappa + \zeta_1^2})} \cdot \log^3(\frac{BMd}{\rho\omega(\zeta_2^2\kappa + \zeta_1^2)}).$$

As mentioned before, we can bound the number of occurrences where the $\mathrm{drift}$ gets large and hence bound the total time we query the oracle of the first kind.

**Lemma 3.7.** *Under the event that $\|\nabla_t - \nabla F(x_t)\| \leq \gamma/4$ for all $t \in [T]$ and our parameter settings, letting $K = \{t \in [T] : \mathrm{drift}_t \geq \kappa\}$ be the set of iterations where the drift is large, we know $|K| \leq O\left(\frac{B\eta}{\kappa} + T\gamma^2\eta^2/\kappa\right) = O\left(B\eta\log^4(\frac{dMB}{\rho\gamma\omega})/\kappa\right)$.*

# 4 Private SOSP

We adopt the framework before and get our main results on finding SOSP privately by constructing private gradient oracles in this section. Finding SOSP for empirical risk function $F_\mathcal{D}$ and for population risk function $F_\mathcal{P}$ are discussed in Subsection 4.1 and Subsection 4.2 respectively.

## 4.1 Convergence to the SOSP of the Empirical Risk

We use Stochastic Spider to improve the convergence to $\alpha$-SOSP of the empirical risk, and aim at getting $\alpha = \tilde{O}(d^{1/3}/n^{2/3})$. We use the full-batch size for simplicity, and use the gradient oracles

$$\mathcal{O}_1(x) := \nabla F_\mathcal{D}(x) + g_1, \quad \text{and} \quad \mathcal{O}_2(x, y) := \nabla F_\mathcal{D}(x) - \nabla F_\mathcal{D}(y) + g_2, \tag{2}$$

where $g_1 \sim \mathcal{N}(0, \sigma_1^2 I_d)$ and $g_2 \sim \mathcal{N}(0, \sigma_2^2\|x - y\|_2^2 I_d)$ are added to ensure privacy by Gaussian mechanism (in Appendix).

Before stating the formal results, note that by Lemma 3.6, the framework can only guarantee the existence of an $\alpha$-SOSP in the outputted set. In order to find the SOSP privately from the set, we adopt the well-known AboveThreshold algorithm, whose pseudo-code can be found in Algorithm 2 in the Appendix. Algorithm 2 is a slight modification of the well-known AboveThreshold algorithm in [20], and we get the following guarantee immediately.

**Lemma 4.1.** *Algorithm 2 is $(\varepsilon, 0)$-DP. Given the point set $\{x_1, \cdots, x_T\}$ and $S$ of size $n$ as the input, (i) if it outputs any point $x_i$, then with probability at least $1 - \omega$, we know*

$$\|\nabla F_S(x_i)\| \leq \alpha + \frac{32 \log(2T/\omega)G}{n\varepsilon}, \text{ and } \text{smin}(\nabla^2 F_S(x_i)) \geq -\sqrt{\rho\alpha} - \frac{32 \log(2T/\omega)M}{n\varepsilon}$$

*(ii) if there exists a $\alpha$-SOSP point $x \in \{x_i\}_{i \in [T]}$, then with probability at least $1 - \omega$, Algorithm 2 will output one point.*

Choosing the appropriate noise scales for the Gaussian added in Equation (2) and running Algorithm 1 can get a private set of points which contains at least one good SOSP. Then we can run Algorithm 2 to find the good SOSP in the set privately. The formal guarantee is stated below:

**Theorem 4.2** (Empirical). *For $\varphi = O(1)$, use Equation (2) as gradient oracles with $\kappa = \frac{G^{4/3}B^{1/3}}{M^{5/3}}(\frac{\sqrt{d}}{n\sqrt{\varphi}})^{2/3}$, $\sigma_1 = \frac{G\sqrt{B\eta/\kappa}\log^2(dMB/\omega)}{n\sqrt{\varphi}}, \sigma_2 = \frac{M\sqrt{BM/\alpha_1^2}\log^5(dMB/\omega)}{n\sqrt{\varphi}}$. Running Algorithm 1, outputting the set $\{x_i\}_{i \in [T]}$ if the total time to query $\mathcal{O}_1$ is bounded by $O(B\eta \log^4(\frac{dMB}{\rho\gamma\omega})/\kappa)$, otherwise outputting a set of $T$ arbitrary points is $(\varphi/2)$-zCDP. With probability at least $1 - \omega$, at least one point in the output set is $\alpha_1$-SOSP of $F_{\mathcal{D}}$ with*

$$\alpha_1 = O\left(\left(\frac{\sqrt{dBGM}}{n\sqrt{\varphi}}\right)^{2/3} \cdot \log^6\left(\frac{nBMd}{\rho\omega}\right)\right).$$

*Moreover, if we run Algorithm 2 with inputs $\{x_i\}_{i \in [T]}, \mathcal{D}, B, M, G, \rho, \alpha_1$, and $\varepsilon = \sqrt{\varphi}$, with probability at least $1 - \omega$, we can get an $\alpha_2$-SOSP of $F_{\mathcal{D}}$ with $\alpha_2 = O\left(\alpha_1 + \frac{G\log(n/G\omega)}{n\sqrt{\varphi}} + \frac{M\log(dBGM/\rho\omega)}{n\sqrt{\varphi}\sqrt{\rho}}\sqrt{\alpha_1}\right)$. The whole procedure is $\varphi$-zCDP.*

For more generality, we state the above theorem in terms of zCDP. $\varphi$-zCDP implies $(\varphi + 2\sqrt{\varphi \log(1/\delta)}, \delta)$-DP, which gives the bounds in Table 1 (see Appendix for details).

**Remark 4.3.** *It's worth noting that the cost of gradient computation can be reduced by utilizing smaller batch sizes. However, our work does not focus on optimizing this aspect.*

## 4.2 Convergence to the SOSP of the Population Risk

This subsection aims at getting an $\alpha$-SOSP for $F_{\mathcal{P}}$ (the population function). Differing from the stochastic oracles used for empirical function $F_{\mathcal{D}}$, we do not use full batch in the oracle. As an alternative, we draw fresh samples from $\mathcal{D}$ without replacement with a smaller batch size:

$$\mathcal{O}_1(x) := \frac{1}{b_1}\sum_{z \in S_1}\nabla f(x; z) + g_1, \text{ and } \mathcal{O}_2(x, y) := \frac{1}{b_2}\sum_{z \in S_2}(\nabla f(x; z) - \nabla f(y; z)) + g_2, \quad (3)$$

where $S_1$ and $S_2$ are sets of size of $b_1$ and $b_2$ respectively drawn from $\mathcal{D}$ without replacement, $g_1 \sim \mathcal{N}(0, \sigma_1^2 I_d)$ and $g_2 \sim \mathcal{N}(0, \sigma_2^2\|x - y\|_2^2 \cdot I_d)$ are added for privacy guarantee. These gradient oracles satisfy the following.

**Claim 4.4.** *The gradient oracles $\mathcal{O}_1$ and $\mathcal{O}_2$ constructed in Equation (3) are a first kind of $O(\frac{L\sqrt{\log d}}{\sqrt{b_1}} + \sqrt{d}\sigma_1)$ norm-subGaussian gradient oracle and second kind of $O(\frac{M\sqrt{\log d}}{\sqrt{b_2}} + \sqrt{d}\sigma_2)$ norm-subGaussian gradient oracle respectively.*

Recall that in the empirical case, we use Algorithm 2 to choose the SOSP for $F_{\mathcal{D}}$. But in the population case, we need to find SOSP for $F_{\mathcal{P}}$, and what we have are samples from $\mathcal{P}$. We need the following technical results to help us find the SOSP from the set, which follows from Hoeffding inequality for norm-subGaussians (Lemma 2.4) and Matrix Bernstein inequality (in the Appendix).

**Lemma 4.5.** *Fix a point $x \in \mathbb{R}^d$. Given a set $S$ of $m$ samples drawn i.i.d. from the distribution $\mathcal{P}$, then we know with probability at least $1 - \omega$, we have*

$$\|\nabla F_S(x) - \nabla F_{\mathcal{P}}(x)\|_2 \leq O\big(\frac{G \log(d/\omega)}{\sqrt{m}}\big) \bigwedge \|\nabla^2 F_S(x) - \nabla^2 F_{\mathcal{P}}(x)\|_{op} \leq O\big(\frac{M \log(d/\omega)}{\sqrt{m}}\big).$$

By choosing the appropriate noise scales $\sigma_1$ and $\sigma_2$ to ensure the privacy guarantee, we can bound the population bound similar to the empirical bound with these tools.

**Theorem 4.6** (Population). *Divide the dataset $\mathcal{D}$ into two disjoint datasets $\mathcal{D}_1$ and $\mathcal{D}_2$ of size $\lceil n/2 \rceil$ and $\lfloor n/2 \rfloor$ respectively. Set $b_1 = \frac{n\kappa}{B\eta}, b_2 = \frac{n\alpha_1^2}{BM}, \sigma_1 = \frac{G}{b_1\sqrt{\varphi}}, \sigma_2 = \frac{M}{b_2\sqrt{\varphi}}$ and $\kappa = \max(\frac{G^{4/3}B^{1/3}\log^{1/3} d}{M^{5/3}}n^{-1/3}, (\frac{GB^{2/3}}{M^{5/3}})^{6/7}(\frac{\sqrt{d}}{n\sqrt{\varphi}})^{4/7})$ in Equation (3) and use them as gradient oracles. Running Algorithm 1 with $\mathcal{D}_1$, and outputting the set $\{x_i\}_{i \in [T]}$ if the total time to query $\mathcal{O}_1$ is bounded by $O(B\eta \log^4(\frac{dMB}{\rho\gamma\omega})/\kappa)$, otherwise outputting a set of $T$ arbitrary points, is $(\varphi/2)$-zCDP, and with probability at least $1 - \omega$, at least one point in the output is $\alpha_1$-SOSP of $F_{\mathcal{P}}$ with*

$$\alpha_1 = O\Big(\big((BGM \cdot \log d)^{1/3}\frac{1}{n^{1/3}} + (G^{1/7}B^{3/7}M^{3/7})(\frac{\sqrt{d}}{n\sqrt{\varphi}})^{3/7}\big)\log^3(nBMd/\rho\omega)\Big).$$

*Moreover, if we run Algorithm 2 with inputs $\{x_i\}_{i \in [T]}, \mathcal{D}_2, B, M, G, \rho, \alpha_1, \varepsilon = \sqrt{\varphi}$ with probability at least $1 - \omega$, Algorithm 2 can output an $\alpha_2$-SOSP of $F_{\mathcal{P}}$ with $\alpha_2 = O\left(\alpha_1 + \frac{M \log(ndBGM/\rho\omega)}{\sqrt{\rho}\min(n\sqrt{\varphi}, n^{1/2})}\sqrt{\alpha_1} + G(\frac{\log(n/G\omega)}{n\sqrt{\varphi}} + \frac{\log(d/\omega)}{\sqrt{n}})\right)$. The whole procedure is $\varphi$-zCDP.*

## 5 Bounding the Private Excess Risk

In this section, we shift our focus from "second-order" guarantees to "zeroth-order" guarantees, and consider the problem of getting good private risk bounds without convexity.

### 5.1 Polynomial Time Approach

If we want the algorithm to be efficient and implementable in polynomial time, to our knowledge the only known bound is $O(\frac{d \log(1/\delta)}{\varepsilon^2 \log n})$ in [46] for smooth functions. [46] used Gradient Langevin Dynamics, a popular variant of SGD to solve this problem, and prove the privacy by advanced composition [31]. We generalize the exponential mechanism to the non-convex case and implement it without a smoothness assumption.

First recall the Log-Sobolev inequality: We say a probability distribution $\pi$ satisfies LSI with constant $C_{\text{LSI}}$ if for all $f : \mathbb{R}^d \to \mathbb{R}, \mathbb{E}_\pi[f^2 \log f^2] - \mathbb{E}_\pi[f^2] \log \mathbb{E}_\pi[f^2] \leq 2C_{\text{LSI}} \mathbb{E}_\pi \|\nabla f\|_2^2$. A well-known result ([40]) says if $f$ is $\mu$-strongly convex, then the distribution proportional to $\exp(-f)$ satisfies LSI with constant $1/\mu$. Recall the results from previous results [39] about LSI and DP:

**Theorem 5.1** ([39]). *Sampling from $\exp(-\beta F(x; \mathcal{D}) - r(x))$ for some public regularizer $r$ is $(\varepsilon, \delta)$-DP, where $\varepsilon \leq 2\frac{G\beta}{n}\sqrt{C_{\text{LSI}}}\sqrt{1 + 2\log(1/\delta)}$, and $C_{\text{LSI}}$ is the worst LSI constant.*

We can apply the classic perturbation lemma to get the new LSI constant in the non-convex case. Suppose we add a regularizer $\frac{\mu}{2}\|x\|^2$, and try to sample from $\exp(-\beta(F(x; \mathcal{D}) + \frac{\mu}{2}\|x\|^2))$.

**Lemma 5.2** (Stroock perturbation). *Suppose $\pi$ satisfies LSI with constant $C_{\text{LSI}}(\pi)$. If $0 < c \leq \frac{d\pi'}{d\pi} \leq C$, then $C_{\text{LSI}}(\pi') \leq \frac{C}{c}C_{\text{LSI}}(\pi)$.*

Lemma 5.3 is a more general version of Theorem 3.4 in [23] and can be used to bound the empirical risk.

**Lemma 5.3.** *Let $\pi(x) \propto \exp(-\beta(F_{\mathcal{D}}(x) + \frac{\mu}{2}\|x\|_2^2))$. Then for $\beta GD > d$, we know*

$$\mathbb{E}_{x \sim \pi}(F_{\mathcal{D}}(x) + \frac{\mu}{2}\|x\|_2^2) - \min_{x^* \in \mathcal{K}}(F_{\mathcal{D}}(x^*) + \frac{\mu}{2}\|x^*\|_2^2) \leq \frac{d}{\beta}\log(\beta GD/d)$$

We now turn to bound the generalization error, and use the notion of uniform stability:

**Lemma 5.4** (Stability and Generalization [10]). *Given a dataset $\mathcal{D} = \{s_i\}_{i \in [n]}$ drawn i.i.d. from some underlying distribution $\mathcal{P}$, and given any algorithm $\mathcal{A}$, suppose we randomly replace a sample $s$ in $\mathcal{D}$ by an independent fresh one $s'$ from $\mathcal{P}$ and get the neighboring dataset $\mathcal{D}'$, then $\mathbb{E}_{\mathcal{D},\mathcal{A}}[F_{\mathcal{P}}(\mathcal{A}(\mathcal{D})) - F_{\mathcal{D}}(\mathcal{A}(\mathcal{D}))] = \mathbb{E}_{\mathcal{D},s',\mathcal{A}}[f(\mathcal{A}(\mathcal{D}); s')) - f(\mathcal{A}(\mathcal{D}'); s'))]$, where $\mathcal{A}(\mathcal{D})$ is the output of $\mathcal{A}$ with input $\mathcal{D}$.*

As each function $f(; s')$ is $G$-Lipschitz, it suffices to bound the $W_2$ distance of $\mathcal{A}(\mathcal{D})$ and $\mathcal{A}(\mathcal{D}')$. If $\mathcal{A}$ is sampling from the exponential mechanism, letting $\pi_{\mathcal{D}} \propto \exp(-\beta(F_{\mathcal{D}}(x) + \frac{\mu}{2}\|x\|^2))$ and $\pi_{\mathcal{D}'} \propto \exp(-\beta(F_{\mathcal{D}'}(x) + \frac{\mu}{2}\|x\|^2))$, it suffices to bound the $W_2$ distance between $\pi_{\mathcal{D}}$ and $\pi_{\mathcal{D}'}$. The following lemma can bound the generalization risk of the exponential mechanism under LSI:

**Lemma 5.5** (Generalization error bound). *Let $\pi_{\mathcal{D}} \propto \exp(-\beta(F_{\mathcal{D}}(x) + \frac{\mu}{2}\|x\|_2^2))$. Then we have $\mathbb{E}_{\mathcal{D}, x \sim \pi_{\mathcal{D}}}[F_{\mathcal{P}}(x) - F_{\mathcal{D}}(x)] \leq O(\frac{G^2 \exp(\beta GD)}{n\mu})$.*

We get the following results:

**Theorem 5.6** (Risk bound). *We are given $\varepsilon, \delta \in (0, 1/2)$. Sampling from $\exp(-\beta(F_{\mathcal{D}}(x) + \frac{\mu}{2}\|x\|_2^2))$ with $\beta = O(\frac{\varepsilon \log(nd)}{GD\sqrt{\log(1/\delta)}}), \mu = \frac{d}{D^2\beta}$ is $(\varepsilon, \delta)$-DP. The empirical risk and population risk are bounded by $O(GD\frac{d \cdot \log\log(n)\sqrt{\log(1/\delta)}}{\varepsilon \log(nd)})$.*

**Implementation**    There are multiple existing algorithms that can sample efficiently from density with LSI, under mild assumptions. For example, when the functions are smooth or weakly smooth, one can turn to the Langevin Monte Carlo [16], and [36]. The algorithm in [46] also requires mild smoothness assumptions. We discuss the implementation of non-smooth functions in bit more details, which is more challenging.

We can adopt the rejection sampler in [26], which is based on the alternating sampling algorithm in [35]. Both [35] and [26] are written in the language of log-concave and strongly log-concave densities, but their results hold as long as LSI holds. By combining them together, we can get the following risk bounds. The details of the implementation can be found in Appendix D.3.

**Theorem 5.7** (Implementation, risk bound). *For $\varepsilon, \delta \in (0, 1/2)$, there is an $(\varepsilon, 2\delta)$-DP efficient sampler that can achieve the empirical and population risks $O(GD\frac{d \cdot \log\log(n)\sqrt{\log(1/\delta)}}{\varepsilon \log(nd)})$. Moreover, in expectation, the sampler takes $\tilde{O}\left(n\varepsilon^3 \log^3(d)\sqrt{\log(1/\delta)}/(GD)\right)$ function values query and some Gaussian random variables restricted to the convex set $\mathcal{K}$ in total.*

## 5.2    Exponential Time Approach

In [23], it is shown that sampling from $\exp(-\frac{\varepsilon n}{GD}F_{\mathcal{D}}(x))$ is $\varepsilon$-DP, and a nearly tight empirical risk bound of $\tilde{O}(\frac{DGd}{n\varepsilon})$ is achieved for convex functions. It is open what is the bound we can get for non-convex DP-SO.

**Upper Bound**    Given exponential time we can use a discrete exponential mechanism as considered in [9]. We recap the argument and extend it to DP-SO. The proof is based on a simple packing argument, and can be found in the Appendix.

**Theorem 5.8.** *There exists an $\varepsilon$-DP differentially private algorithm that achieves a population risk of $O\left(GD\left(d\log(\varepsilon n/d)/(\varepsilon n) + \sqrt{d\log(\varepsilon n/d)}/(\sqrt{n})\right)\right)$.*

**Lower Bound**    Results in [23] imply that the first term of $\tilde{O}(GDd/\varepsilon n)$ is tight, even if we relax to approximate DP with $\delta > 0$. A reduction from private selection problem shows the $\tilde{O}(\sqrt{d/n})$ generalization term is also nearly-tight (Theorem 5.11). In the selection problem, we have $k$ coins, each with an unknown probability $p_i$. Each coin is flipped $n$ times such that $\{x_{i,j}\}_{j \in [n]}$, each $x_{i,j}$ i.i.d. sampled from $\text{Bern}(p_i)$, and we want to choose a coin $i$ with the smallest $p_i$. The risk of choosing $i$ is $p_i - \min_{i^*} p_{i^*}$.

**Theorem 5.9.** *Any algorithm for the selection problem has excess population risk $\tilde{\Omega}(\sqrt{\frac{\log k}{n}})$.*

This follows from a folklore result on the selection problem (see e.g. [5]). We can combine this with the following reduction from selection to non-convex optimization:

**Theorem 5.10** (Restatement of results in [23]). *If any $(\varepsilon, \delta)$-DP algorithm for selection has risk $R(k)$, where $R(k)$ is a function with $k$ as variables, then any $(\varepsilon, \delta)$-DP algorithm for minimizing 1-Lipschitz losses over $B_d(0, 1)$ (the d-dimensional unit ball) has risk $R(2^{\Theta(d)})$.*

From this and the aforementioned lower bounds in empirical non-convex optimization we get the following:

**Theorem 5.11.** *For $\varepsilon \leq 1, \delta \in [2^{-\Omega(n)}, 1/n^{1+\Omega(1)}]$, any $(\varepsilon, \delta)$-DP algorithm for minimizing 1-Lipschitz losses over $B_d(0, 1)$ has excess population risk $\max\{\Omega(d\log(1/\delta)/(\varepsilon n)), \tilde{\Omega}(\sqrt{d/n})\}$.*

## 6 Discussion

In this paper, we gave improved bounds for finding SOSPs under differential privacy, as well as points with low population risk. We discuss some potential follow-up questions here.

First, there is still a gap between our upper bounds and the lower bounds of [2]. Closing this gap is an interesting question and may lead to novel technical insights about private optimization. To obtain upper bounds for finding SOSP, both our work and the work of [2] uses the oracle of the second kind defined in Definition 3.2. Second, in typical private optimization work, only oracles of the first kind are used. It is possible that the oracle of the second kind is useful in private optimization problems besides finding stationary points, either in theory or practice. In particular, since this oracle has lower sensitivity, it may allow us to tolerate a higher noise level / lower privacy budget across multiple iterations. Third, we privatize the SpiderBoost in the work, and there are other versions of variance-reduced algorithms like Spiker and SARAH. It is interesting if our ideas can be used to privatize those algorithms, and compare their practical performance. Lastly, our polynomial-time excess population risk bounds have a $O(1/\log n)$ dependence on the dataset size, whereas for convex losses standard results have a $\max\{1/\sqrt{n}, 1/\varepsilon n\}$ dependence. The stronger dependence is achievable under LSI, but the practical settings in which LSI holds without convexity holding seems limited. It remains an open question to find a practical assumption weaker than convexity that allows us to achieve better dependence on $n$.

## 7 Acknowledgement

DG would like to thank Ruoqi Shen and Kevin Tian for several discussions.

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

# A    Other Preliminary

**Definition A.1** (Laplace distribution). *We say $X \sim \text{Lap}(b)$ if $X$ has density $f(X = x) = \frac{1}{2b}\exp(\frac{-|x|}{b})$.*

In the analysis we use zero-concentrated differential privacy (zCDP):

**Definition A.2** (zCDP). *A mechanism $M$ is $\varphi$-zCDP if for all $\alpha > 1$ and neighboring $\mathcal{D}, \mathcal{D}'$:*

$$R_\alpha(M(\mathcal{D}), M(\mathcal{D}')) \leq \varphi\alpha,$$

*where $R_\alpha$ denotes the $\alpha$-Rényi divergence.*

**Theorem A.3** (zCDP composition, [11]). *The (adaptive) composition of $k$ mechanisms that satisfy $\varphi_1$-zCDP, $\varphi_2$-zCDP, ..., $\varphi_k$-zCDP respectively is $(\varphi_1 + \varphi_2 + \ldots + \varphi_k)$-zCDP.*

**Lemma A.4.** *[Conversion between $(\varepsilon, \delta)$-DP and zCDP, [11]] A mechanism that is $(\varepsilon, 0)$-DP is also $\varepsilon^2/2$-zCDP. A mechanism that is $\varphi$-zCDP is also $(\varepsilon, \delta)$-DP for any $\delta > 0$ and $\varepsilon = \varphi + 2\sqrt{\varphi \log(1/\delta)}$.*

**Theorem A.5** (zCDP of Gaussian Mechanism, [11]). *Given a randomized algorithm $\mathcal{A} : P^* \to \mathbb{R}^d$, let $\Delta_2 f = \max_{neighboring\ \mathcal{D}, \mathcal{D}'} \|\mathcal{A}(\mathcal{D}) - \mathcal{A}(\mathcal{D}')\|_2$, then releasing $\mathcal{A}(\mathcal{D})$ plus noise $\mathcal{N}(0, \sigma^2)$ is $\frac{(\Delta_2 f)^2}{2\sigma^2}$-zCDP.*

**Theorem A.6** (Matrix Bernstein inequality, [45]). *Consider a sequence $\{X_i\}_{i \in m}$ of independent, mean-zero, symmetric $d \times d$ random matrices. If for each matrix $X_i$, we know $\|X_i\|_{op} \leq M$, then for all $t \geq 0$, we have $\Pr\left[\|\sum_{i \in [m]} X_i\|_{op} \geq t\right] \leq d\exp\left(\frac{-t^2}{2(\sigma^2 + Mt/3)}\right)$, where $\sigma^2 = \|\sum_{i \in [m]} \mathbb{E} X_i^2\|_{op}$.*

# B    Omitted Proof of Section 3

## B.1    Proof of Lemma 3.3

**Lemma 3.3.** *For any $0 \leq t \leq T$ and letting $\tau_t \leq t$ be the largest integer such that $\text{drift}_{\tau_t}$ is set to be 0, with probability at least $1 - \omega/T$, for some universal constant $C > 0$, we have*

$$\|\nabla_t - \nabla F(x_t)\|^2 \leq \left(\zeta_2^2 \cdot \sum_{i=\tau_t+1}^{t} \|x_i - x_{i-1}\|^2 + 4\zeta_1^2\right) \cdot C \cdot \log(Td/\omega). \tag{1}$$

*Hence with probability at least $1 - \omega$, we know for each $t \leq T$, $\|\nabla_t - \nabla F(x_t)\|^2 \leq \gamma^2/16$, where $\gamma^2 := 16C(\zeta_2^2 \kappa + 4\zeta_1^2) \cdot \log(Td/\omega)$ and $\kappa$ is a parameter we can choose in the algorithm.*

*Proof.* If $\text{drift}_{\tau_t} = 0$ happens, we use the first kind oracle to query the gradient, and hence $\nabla_{\tau_t} - \nabla F(x_{\tau_t})$ is zero-mean and $\text{nSG}(2\zeta_1)$. If $t = \tau_t$, Equation (1) holds by the property of norm-subGaussian.

For each $\tau_t + 1 \leq i \leq t$, conditional on $\nabla_{i-1}$, we know $\Delta_i - (\nabla F(x_i) - F(x_{i-1}))$ is zero-mean and $\text{nSG}(\zeta_2\|x_i - x_{i-1}\|)$. Note that

$$\nabla_t - \nabla F(x_t) = \nabla_{\tau_t} - \nabla F(x_{\tau_t}) + \sum_{i=\tau_t+1}^{t} [\Delta_i - (\nabla F(x_i) - \nabla F(x_{i-1}))].$$

Equation (1) follows from Lemma 2.4.

We know $\text{drift}_{t-1} = \sum_{i=\tau_t+1}^{t} \|x_i - x_{i-1}\|^2 \leq \kappa$ almost surely by the design of the algorithm. By union bound, we know with probability at least $1 - \omega$, for each $t \in [T]$,

$$\|\nabla_t - \nabla F(x_t)\|^2 \leq C(\zeta_2^2 \kappa + 4\zeta_1^2) \cdot \log(Td/\omega) = \gamma^2/16.$$

$\square$

## B.2 Discussion of Lemma 3.4

**Lemma 3.4** (Essentially from [46]). *Under Assumption 3.1, run SGD iterations $x_{t+1} = x_t - \eta \nabla_t$, with step size $\eta = 1/M$. Suppose $x_0$ is a saddle point satisfying $\|\nabla F(x_0)\| \leq \alpha$ and $\mathrm{smin}(\nabla^2 F(x_0)) \leq -\sqrt{\rho \alpha}$, $\alpha = \gamma \log^3(dBM/\rho \omega)$. If $\nabla_0 = \nabla F(x_0) + \zeta_1 + \zeta_2$ where $\|\zeta_1\| \leq \gamma$, $\zeta_2 \sim \mathcal{N}(0, \frac{\gamma^2}{d \log(d/\omega)} I_d)$, and $\|\nabla_t - \nabla F(x_t)\| \leq \gamma$ for all $t \in [\Gamma]$, with probability at least $1 - \omega \cdot \log(1/\omega)$, one has $F(x_\Gamma) - F(x_0) \leq -\Omega\big(\frac{\gamma^{3/2}}{\sqrt{\rho} \log^3(\frac{dMB}{\rho \gamma \omega})}\big)$, where $\Gamma = \frac{M \log(\frac{dMB}{\rho \gamma \omega})}{\sqrt{\rho \gamma}}$.*

We briefly recap the proof of Lemma 3.4 in [46]. One observation between the decreased function value, and the distance solutions moved is stated below:

**Lemma B.1** (Lemma 11, [46]). *For each $t \in [\Gamma]$, we know*

$$\|x_{t+1} - x_0\|_2^2 \leq 8\eta(\Gamma(F(x_0) - F(x_\Gamma)) + 50\eta^2 \Gamma \sum_{i \in [\Gamma]} \|\nabla_i - \nabla F(x_t)\|_2^2.$$

The difference between our algorithm and the DP-GD in [46] is the noise on the gradient. Note that with high probability, $\sum_{i \in [\Gamma]} \|\nabla_i - \nabla F(x_t)\|_2^2$ in our algorithm is controlled and small, and hence does not change the other proofs in [46]. Hence if $F(x_0) - F(x_\Gamma)$ is small, i.e., the function value does not decrease significantly, we know $x_t$ is close to $x_0$.

Let $B_x(r)$ be the unit ball of radius $r$ around point $x$. Denote the $(x)_\Gamma$ the point $x_\Gamma$ after running SGD mentioned in Lemma 3.4 for $\Gamma$ steps beginning at $x$. With this observation, denote $B^\gamma(x_0) := \{x \mid x \in B_{x_0}(\eta \alpha), \Pr[F((x)_\Gamma) - F(x) \geq -\Phi] \geq \omega\}$. [46] demonstrates the following lemma:

**Lemma B.2.** *If $\|\nabla F(x_0)\| \leq \alpha$ and $\mathrm{smin}(\nabla^2 F(x_0)) \leq -\sqrt{\rho \gamma}$, then the width of $B^\gamma(x_0)$ along the along the minimum eigenvector of $\nabla^2 F(x_0)$ is at most $\frac{\omega \eta \gamma}{\log(1/\omega)} \sqrt{\frac{2\pi}{d}}$.*

The intuition is that if two different points $x^1, x^2 \in B_{x_0}(\eta \alpha)$, and $x^1 - x^2$ is large along the minimum eigenvector, then with high probability, the distance between $\|(x^1)_\Gamma - (x^2)_\Gamma\|$ will be large, and either $\|(x^1)_\Gamma - x^1\|$ or $\|(x^2)_\Gamma - x^2\|$ is large, and hence either $F(x^1) - F((x^1)_\Gamma)$ or $F(x^2) - F((x^2)_\Gamma)$ is large. The Lemma 3.4 follows from Lemma B.2 by using the Gaussian $\zeta_2$ to kick off the point.

## B.3 Proof of Lemma 3.5

**Lemma 3.5.** *By setting $\eta = 1/M$, we have $F(x_{t+1}) \leq F(x_t) + \eta \|\nabla_t\| \cdot \|\nabla F(x_t) - \nabla_t\| - \frac{\eta}{2} \|\nabla_t\|^2$. Moreover, with probability at least $1 - \omega$, for each $t \leq T$ such that $\|\nabla F(x_t)\| \geq \gamma$, we have*

$$F(x_{t+1}) - F(x_t) \leq -\eta \|\nabla_t\|^2/6 \leq -\eta \gamma^2/6.$$

*Proof.* By the assumption on smoothness, we know

$$F(x_{t+1}) \leq F(x_t) + \langle \nabla F(x_t), x_{t+1} - x_t \rangle + \frac{M}{2} \|x_{t+1} - x_t\|^2$$

$$= F(x_t) - \eta/2 \|\nabla_t\|^2 - \langle \nabla F(x_t) - \nabla_t, \eta \nabla_t \rangle$$

$$\leq F(x_t) + \eta \|\nabla F(x_t) - \nabla_t\| \cdot \|\nabla_t\| - \frac{\eta}{2} \|\nabla_t\|^2.$$

By Lemma 3.3, with probability at least $1 - \omega$, for each $t \in [T]$ we have $\|\nabla F(x_t) - \nabla_t\|_2 \leq \gamma/4$. Hence we know if $\nabla F(x_t) \geq \gamma$, we have

$$F(x_{t+1}) - F(x_t) \leq -\eta \|\nabla_t\|^2/6 \leq -\eta \gamma^2/6.$$

$\square$

## B.4 Proof of Lemma 3.6

**Lemma 3.6.** *Suppose $\mathcal{O}_1$ and $\mathcal{O}_2$ are $\zeta_1$ and $\zeta_2$ norm-subGaussian respectively. If one sets $\gamma = O(1)\sqrt{(\zeta_2^2 \kappa + 4\zeta_1^2) \cdot \log(Td/\omega)}$, with probability at least $1 - \omega$, at least one point in the output set*

$\{x_1, \cdots, x_T\}$ of Algorithm 1 is $\alpha$-SOSP, where

$$\alpha = \gamma \log^3(BMd/\rho\omega\gamma) = \sqrt{(\zeta_2^2\kappa + 4\zeta_1^2) \cdot \log(\frac{d/\omega}{\zeta_2^2\kappa + \zeta_1^2})} \cdot \log^3(\frac{BMd}{\rho\omega(\zeta_2^2\kappa + \zeta_1^2)}).$$

*Proof.* By Lemma 3.5, we know if the gradient $\|\nabla F(x_t)\| \geq \gamma$, then with high probability that $F(x_{t+1}) - F(x_t) \leq -\eta\gamma^2/6$. By Lemma 3.4, if $x_t$ is a saddle point (with small gradient norm but the Hessian has a small eigenvalue), then with high probability that $F(x_{\Gamma+t}) - F(x_t) \leq -\Omega(\frac{\gamma^{3/2}}{\sqrt{\rho}\log^3(\frac{dMB}{\rho\gamma\omega})})$, and the function values decrease $\Omega(\frac{\gamma^2}{M\log^4(\frac{dMB}{\rho\gamma\omega})})$ on average for each step.

Recall the assumption that the risk is upper bounded by $B$, by our setting $T = \Omega(\frac{BM}{\gamma^2}\log^4(\frac{dMB}{\rho\gamma\omega}))$, the statement is proved. $\square$

### B.5 Proof of Lemma 3.7

**Lemma 3.7.** *Under the event that $\|\nabla_t - \nabla F(x_t)\| \leq \gamma/4$ for all $t \in [T]$ and our parameter settings, letting $K = \{t \in [T] : \text{drift}_t \geq \kappa\}$ be the set of iterations where the drift is large, we know $|K| \leq O(\frac{B\eta}{\kappa} + T\gamma^2\eta^2/\kappa) = O(B\eta\log^4(\frac{dMB}{\rho\gamma\omega})/\kappa).$*

*Proof.* By Lemma 3.5, if $\|F(x_t)\| \geq \gamma$, we know $F(x_{t+1}) - F(x_t) \leq -\eta\|\nabla_t\|^2/6$, and $F(x_{t+1}) - F(x_t) \leq \eta\gamma^2$ otherwise. Index the items in $K = \{t_1, t_2, \cdots, t_{|K|}\}$ such that $t_i < t_{i+1}$. We know

$$F(x_{t_{i+1}}) - F(x_{t_i}) \leq -\frac{1}{6\eta}\text{drift}_{t_{i+1}} + (t_{i+1} - t_i)\gamma^2\eta \leq -\frac{1}{6\eta}\kappa + (t_{i+1} - t_i)\gamma^2\eta.$$

Recall by the assumption that $\max_y F(y) - \min_x F(x) \leq B$. And hence $-B \leq F(x_{t_{|L|}}) - F(x_{t_1}) \leq -\frac{|K|}{6\eta}\kappa + T\gamma^2\eta$, and we know

$$|K| \leq O(\frac{B\eta}{\kappa} + T\gamma^2\eta^2/\kappa) = O(B\eta\log^4(\frac{dMB}{\rho\gamma\omega})/\kappa).$$

$\square$

## C   Appendix for Section 4

The pseudocode of Algorithm 2 is stated below:

---
**Algorithm 2** AboveThreshold
---
1: **Input:** A set of points $\{x_i\}_{i=1}^T$, dataset $S$, parameters of objective function $B, M, G, \rho$, objective error $\alpha$
2: Set $\widehat{T}_1 = \alpha + \text{Lap}(\frac{4G}{n\varepsilon}) + \frac{16\log(2T/\omega)G}{n\varepsilon}, \widehat{T}_2 = -\sqrt{\rho\alpha} + \text{Lap}(\frac{4M}{n\varepsilon}) - \frac{16\log(2T/\omega)M}{n\varepsilon}$
3: **for** $i = 1, \cdots, T$ **do**
4:   **if** $\|\nabla F_S(x_i)\| + \text{Lap}(\frac{8G}{n\varepsilon}) \leq \widehat{T}_1 \bigwedge \text{smin}(\nabla^2 F_S(x_i)) + \text{Lap}(\frac{8M}{n\varepsilon}) \geq \widehat{T}_2$ **then**
5:     **Output:** $x_i$
6:     **Halt**
7:   **end if**
8: **end for**

---

### C.1   Proof of Theorem 4.2

**Theorem 4.2** (Empirical). *For $\varphi = O(1)$, use Equation (2) as gradient oracles with $\kappa = \frac{G^{4/3}B^{1/3}}{M^{5/3}}(\frac{\sqrt{d}}{n\sqrt{\varphi}})^{2/3}$, $\sigma_1 = \frac{G\sqrt{B\eta/\kappa}\log^2(dMB/\omega)}{n\sqrt{\varphi}}$, $\sigma_2 = \frac{M\sqrt{BM/\alpha_1^2}\log^5(dMB/\omega)}{n\sqrt{\varphi}}$. Running Algorithm 1, outputting the set $\{x_i\}_{i\in[T]}$ if the total time to query $\mathcal{O}_1$ is bounded by $O(B\eta\log^4(\frac{dMB}{\rho\gamma\omega})/\kappa),$*

*otherwise outputting a set of $T$ arbitrary points is $(\varphi/2)$-zCDP. With probability at least $1 - \omega$, at least one point in the output set is $\alpha_1$-SOSP of $F_{\mathcal{D}}$ with*

$$\alpha_1 = O\left(\left(\frac{\sqrt{dBGM}}{n\sqrt{\varphi}}\right)^{2/3} \cdot \log^6\left(\frac{nBMd}{\rho\omega}\right)\right).$$

*Moreover, if we run Algorithm 2 with inputs $\{x_i\}_{i\in[T]}, \mathcal{D}, B, M, G, \rho, \alpha_1$, and $\varepsilon = \sqrt{\varphi}$, with probability at least $1 - \omega$, we can get an $\alpha_2$-SOSP of $F_{\mathcal{D}}$ with $\alpha_2 = O\left(\alpha_1 + \frac{G\log(n/G\omega)}{n\sqrt{\varphi}} + \frac{M\log(dBGM/\rho\omega)}{n\sqrt{\varphi}\sqrt{\rho}}\sqrt{\alpha_1}\right)$. The whole procedure is $\varphi$-zCDP.*

*Proof.* The privacy guarantee can be proved by zCDP composition (Lemma A.3) Gaussian Mechanism (Theorem A.5) and Lemma 3.7. Specifically, by Assumption 3.1 and our settings of parameters, we know the sensitivity of $\mathcal{O}_1$ and $\mathcal{O}_2$ are bounded by $\frac{G}{n}$ and $\frac{M\|x-y\|}{n}$ respectively, and querying $\mathcal{O}_1$ and $\mathcal{O}_2$ each time are $\frac{G^2}{n^2\sigma_1^2}$-zCDP and $\frac{M^2}{n^2\sigma_2^2}$-zCDP respectively. We can apply the advanced composition to prove the privacy guarantee of the whole algorithm. As the total number of iterations $T$ is fixed, all queries made to $\mathcal{O}_2$ satisfy $\frac{TM^2}{n^2\sigma_2^2}$-zCDP. It suffices to bound the total time to query $\mathcal{O}_1$, which is guaranteed in the statement. That is the total times to query $\mathcal{O}_1$ is $T_1 := O\left(B\eta\log^4(\frac{dMB}{\rho\omega})/\kappa\right)$, so the queries to $\mathcal{O}_1$ satisfy $\frac{T_1G^2}{n^2\sigma_1^2}$-zCDP. If the time exceeds $O\left(B\eta\log^4(\frac{dMB}{\rho\omega})/\kappa\right)$, then we will output a set of arbitrary points which does not occur additional privacy cost. The overall privacy guarantee is $(\frac{G^2T_1}{n^2\sigma_1^2} + \frac{M^2T}{n^2\sigma_2^2})$-zCDP. To get $(\varphi/2)$-zCDP, as desired we can choose:

$$\sigma_1 = \frac{2G\sqrt{T_1}}{n\sqrt{\varphi}} = \frac{2G\log^2(\frac{dMB}{\rho\omega})\sqrt{B\eta/\kappa}}{n\sqrt{\varphi}}, \qquad \sigma_2 = \frac{2M\sqrt{T}}{n\sqrt{\varphi}} = \frac{2M\log^5(\frac{dMB}{\rho\omega})\sqrt{BM}}{n\sqrt{\varphi}\alpha_1}$$

As for the utility, we know the $\mathcal{O}_1$ and $\mathcal{O}_2$ constructed in Equation (2) are first kind of $\sigma_1\sqrt{d}$ and second kind of $\sigma_2\sqrt{d}$ norm-subGaussian gradient oracle by Fact 2.3. Hence by Lemma 3.6, the utility $\alpha_1$ satisfies that

$$\begin{aligned}\alpha_1 &= O(\sigma_1\sqrt{d} + \sigma_2\sqrt{d\kappa}) \cdot \log^3(BMd/\rho\omega) \\ &= O\left(\frac{G\sqrt{dB\eta/\kappa}}{n\sqrt{\varphi}} + \frac{M\log^3(\frac{dMB}{\rho\omega})\sqrt{BM}}{n\sqrt{\varphi}\alpha_1}\sqrt{d\kappa}\right) \cdot \log^5(BMd/\rho\omega).\end{aligned}$$

By Lemma 3.7, with probability at least $1 - \omega$, the total time to query $\mathcal{O}_1$ is controlled and the final output will not be arbitrary points. Choosing the best $\kappa$ demonstrates the bound on $\alpha_1$. The bound for $\alpha_2$ follows from the value of $\alpha_1$ and Lemma 4.1. Combining the two items in Lemma 4.1, we know with probability at least $1 - \omega$, the output point $x$ of Algorithm 2 satisfies that

$$\|\nabla F_{\mathcal{D}}(x)\| \leq \alpha_1 + \frac{32\log(2T/\omega)G}{n\sqrt{\varphi}}, \text{ and } \text{smin}(\nabla^2 F_{\mathcal{D}}(x)) \geq -\sqrt{\rho\alpha_1} - \frac{32\log(2T/\omega)M}{n\sqrt{\varphi}}.$$

Hence we know $x$ is an $\alpha_2$-SOSP for $\alpha_2$ stated in the statement. $\qquad\square$

### C.2 Proof of Claim 4.4

**Claim 4.4.** *The gradient oracles $\mathcal{O}_1$ and $\mathcal{O}_2$ constructed in Equation (3) are a first kind of $O(\frac{L\sqrt{\log d}}{\sqrt{b_1}} + \sqrt{d}\sigma_1)$ norm-subGaussian gradient oracle and second kind of $O(\frac{M\sqrt{\log d}}{\sqrt{b_2}} + \sqrt{d}\sigma_2)$ norm-subGaussian gradient oracle respectively.*

*Proof.* For the oracle $\mathcal{O}_1$, we know for each $z \in S_1$, $\mathbb{E}_{z\sim\mathcal{P}}[\nabla f(x,z)] = \nabla F_{\mathcal{P}}(x)$ and $\nabla f(x,z) - \nabla F_{\mathcal{P}}(x)$ is $\text{nSG}(L)$ due to the Lipschitzness assumption. The statement follows from Fact 2.3 and Lemma 2.4. As for the $\mathcal{O}_2$, the statement follows similarly with the smoothness assumption. $\qquad\square$

## C.3 Proof of Lemma 4.5

**Lemma 4.5.** *Fix a point $x \in \mathbb{R}^d$. Given a set $S$ of $m$ samples drawn i.i.d. from the distribution $\mathcal{P}$, then we know with probability at least $1 - \omega$, we have*

$$\|\nabla F_S(x) - \nabla F_{\mathcal{P}}(x)\|_2 \leq O\big(\frac{G \log(d/\omega)}{\sqrt{m}}\big) \bigwedge \|\nabla^2 F_S(x) - \nabla^2 F_{\mathcal{P}}(x)\|_{op} \leq O\big(\frac{M \log(d/\omega)}{\sqrt{m}}\big).$$

*Proof.* As for any $s \in S$, $\nabla f(x; s) - \nabla F_{\mathcal{P}}(x)$ is zero-mean nSG($G$). Then the Hoeffding inequality for norm-subGuassians (Lemma 2.4) demonstrates with probability at least $1 - \omega/2$, we have $\|\nabla F_S(x) - \nabla F_{\mathcal{P}}(x)\|_2 \leq O\big(\frac{G \log(d/\omega)}{\sqrt{m}}\big)$.

As for the other term, we know for any $s \in S, \mathbb{E}[\nabla^2 f(x; s) - \nabla^2 F_{\mathcal{P}}(x)] = 0$, and $\|\nabla^2 f(x; s) - \nabla^2 F_{\mathcal{P}}(x)\|_{op} \leq 2M$ almost surely. Hence applying Matrix Bernstein inequality (Theorem A.6) with $\sigma^2 = 4M^2m, t = O\big(\sqrt{m}M \log(d/\omega)\big)$, we know with probability at least $1 - \omega/2$, $\|\nabla^2 F_S(x) - \nabla^2 F_{\mathcal{P}}(x)\|_{op} \leq t/m$.

Applying the Union bound completes the proof. $\qquad\square$

## C.4 Proof of Theorem 4.6

**Theorem 4.6** (Population). *Divide the dataset $\mathcal{D}$ into two disjoint datasets $\mathcal{D}_1$ and $\mathcal{D}_2$ of size $\lceil n/2 \rceil$ and $\lfloor n/2 \rfloor$ respectively. Set $b_1 = \frac{n\kappa}{B\eta}, b_2 = \frac{n\alpha_1^2}{BM}, \sigma_1 = \frac{G}{b_1\sqrt{\varphi}}, \sigma_2 = \frac{M}{b_2\sqrt{\varphi}}$ and $\kappa = \max(\frac{G^{4/3}B^{1/3}\log^{1/3}d}{M^{5/3}}n^{-1/3}, (\frac{GB^{2/3}}{M^{5/3}})^{6/7}(\frac{\sqrt{d}}{n\sqrt{\varphi}})^{4/7})$ in Equation (3) and use them as gradient oracles. Running Algorithm 1 with $\mathcal{D}_1$, and outputting the set $\{x_i\}_{i\in[T]}$ if the total time to query $\mathcal{O}_1$ is bounded by $O(B\eta \log^4(\frac{dMB}{\rho\gamma\omega})/\kappa)$, otherwise outputting a set of $T$ arbitrary points, is $(\varphi/2)$-zCDP, and with probability at least $1 - \omega$, at least one point in the output is $\alpha_1$-SOSP of $F_{\mathcal{P}}$ with*

$$\alpha_1 = O\Big(((BGM \cdot \log d)^{1/3}\frac{1}{n^{1/3}} + (G^{1/7}B^{3/7}M^{3/7})(\frac{\sqrt{d}}{n\sqrt{\varphi}})^{3/7}) \log^3(nBMd/\rho\omega)\Big).$$

*Moreover, if we run Algorithm 2 with inputs $\{x_i\}_{i\in[T]}, \mathcal{D}_2, B, M, G, \rho, \alpha_1, \varepsilon = \sqrt{\varphi}$ with probability at least $1 - \omega$, Algorithm 2 can output an $\alpha_2$-SOSP of $F_{\mathcal{P}}$ with $\alpha_2 = O\left(\alpha_1 + \frac{M\log(ndBGM/\rho\omega)}{\sqrt{\rho}\min(n\sqrt{\varphi},n^{1/2})}\sqrt{\alpha_1} + G(\frac{\log(n/G\omega)}{n\sqrt{\varphi}} + \frac{\log(d/\omega)}{\sqrt{n}})\right)$. The whole procedure is $\varphi$-zCDP.*

*Proof.* Recall that we draw the samples to construct the gradient oracles (Equation 3) without replacement, and we should have all samples to be fresh to avoid dependency, and hence we need

$$b_1 \cdot |K| + b_2 \cdot T \leq n/2,$$

which is satisfied by the procedure in the statement, as if the total time to query the $\mathcal{O}_1$ exceeds the threshold, the algorithm fails and outputs a set of arbitrary points. As we never reuse a sample, the privacy guarantee follows directly from the Gaussian Mechanism [20]. Specifically, the sensitivity of querying $\mathcal{O}_1$ and $\mathcal{O}_2$ are bounded by $G/b_1$ and $M\|x - y\|/b_2$ respectively, and querying $\mathcal{O}_1$ and $\mathcal{O}_2$ are $(\varphi/2)$-zCDP by Theorem A.5.

The Norm-subGaussian parameters of the oracles follow from Claim 4.4. By lemma 3.6, we have

$$\frac{\alpha_1}{\log^3(nBMd/\rho\omega)}$$
$$=O(\sigma_1\sqrt{d} + \frac{G\sqrt{\log d}}{\sqrt{b_1}} + \sigma_2\sqrt{d\kappa} + \frac{M\sqrt{\kappa \log d}}{\sqrt{b_2}}).$$
$$=O(\frac{GB\eta\sqrt{d}}{n\sqrt{\varphi}\kappa} + \frac{BM^2}{n\sqrt{\varphi}\alpha_1^2}\sqrt{d\kappa} + \frac{G\sqrt{B\eta \log d}}{\sqrt{n\kappa}} + M\sqrt{\kappa}\frac{\sqrt{BM\log d}}{\sqrt{n}\alpha_1}).$$

Setting $\kappa = \max(\frac{G^{4/3}B^{1/3}\log^{1/3}d}{M^{5/3}}(n)^{-1/3}, (\frac{GB^{2/3}}{M^{5/3}})^{6/7}(\frac{\sqrt{d}}{n\sqrt{\varphi}})^{4/7})$, we get

$$\alpha_1 = O\Big(((BGM \log d)^{1/3}\frac{1}{n^{1/3}} + (G^{1/7}B^{3/7}M^{3/7})(\frac{\sqrt{d}}{n\sqrt{\varphi}})^{3/7}) \log^3(nBMd/\rho\omega)\Big).$$

Then we use the other half fresh samples $\mathcal{D}_2$ to find the point in the set by Algorithm 2. By Lemma 4.1 and Lemma 4.5, we know with probability at least $1 - \omega$, for some large enough constant $C > 0$, the output point $x$ of Algorithm 2 satisfies that

$$\|\nabla F_{\mathcal{P}}(x)\|_2 \leq \alpha_1 + G\left(\frac{32 \log(2T/\omega)}{n\sqrt{\varphi}} + \frac{C \log(dn/\omega)}{\sqrt{n}}\right),$$

$$\mathrm{smin}(\nabla^2 F_{\mathcal{P}}(x)) \geq -\sqrt{\rho\alpha_1} - M\left(\frac{32 \log(2T/\omega)}{n\sqrt{\varphi}} + \frac{C \log(dn/\omega)}{\sqrt{n}}\right)$$

Hence we know $x$ is an $\alpha_2$-SOSP for $\alpha_2$ stated in the statement. The privacy guarantee follows from Lemmas A.3, A.4, and 4.1. $\qquad\square$

## D   Omitted proof of Section 5

### D.1   Proof of Lemma 5.5

**Lemma 5.5** (Generalization error bound). *Let $\pi_{\mathcal{D}} \propto \exp(-\beta(F_{\mathcal{D}}(x) + \frac{\mu}{2}\|x\|_2^2))$. Then we have* $\mathbb{E}_{\mathcal{D}, x \sim \pi_{\mathcal{D}}}[F_{\mathcal{P}}(x) - F_{\mathcal{D}}(x)] \leq O(\frac{G^2 \exp(\beta GD)}{n\mu})$.

*Proof.* We know how to bound the KL divergence by LSI:

$$KL(\pi_{\mathcal{D}}, \pi_{\mathcal{D}'}) := \int \log \frac{\mathrm{d}\pi_{\mathcal{D}}}{\mathrm{d}\pi_{\mathcal{D}'}} \mathrm{d}\pi_{\mathcal{D}}$$

$$\leq \frac{C_{\mathrm{LSI}}}{2} \mathop{\mathbb{E}}_{\pi_{\mathcal{D}}} \left\| \nabla \log \frac{\mathrm{d}\pi_{\mathcal{D}}}{\mathrm{d}\pi_{\mathcal{D}'}} \right\|_2^2$$

$$\leq 2C_{\mathrm{LSI}} G^2 \beta^2 / n^2.$$

LSI can lead to Talagrand transportation inequality [Theorem 1 in [40]], i.e.,

$$W_2(\pi_{\mathcal{D}}, \pi_{\mathcal{D}'}) \lesssim \sqrt{C_{\mathrm{LSI}} \cdot KL(\pi_{\mathcal{D}}, \pi_{\mathcal{D}'})} = C_{\mathrm{LSI}} G\beta / n.$$

The generalization error is bounded by $O(C_{\mathrm{LSI}} G^2 \beta / n)$. Using Holley-Stroock perturbation, we know $C_{\mathrm{LSI}}(\pi_{\mathcal{D}}) \leq \frac{\exp(\beta GD)}{\beta\mu}$ and hence the $W_2$ distance between $\pi_{\mathcal{D}}$ and $\pi_{\mathcal{D}'}$ can be bounded by $O(\frac{G \exp(\beta GD)}{n\mu})$. The statement follows the Lipschitzness constant and Lemma 5.4. $\qquad\square$

### D.2   Proof of Theorem 5.6

**Theorem 5.6** (Risk bound). *We are given $\varepsilon, \delta \in (0, 1/2)$. Sampling from $\exp(-\beta(F_{\mathcal{D}}(x) + \frac{\mu}{2}\|x\|_2^2))$ with $\beta = O(\frac{\varepsilon \log(nd)}{GD\sqrt{\log(1/\delta)}}), \mu = \frac{d}{D^2\beta}$ is $(\varepsilon, \delta)$-DP. The empirical risk and population risk are bounded by $O(GD\frac{d \cdot \log \log(n)\sqrt{\log(1/\delta)}}{\varepsilon \log(nd)})$.*

*Proof.* Denote $\pi(x) \propto \exp(-\beta(F_{\mathcal{D}}(x) + \frac{\mu}{2}\|x\|_2^2))$. By Lemma 5.2, we know $C_{\mathrm{LSI}}(\pi) \leq \frac{1}{\beta\mu} \cdot \exp(\beta GD)$. Plugging in the parameters and applying Theorem 5.1, we get

$$\frac{2G\beta}{n} \cdot \sqrt{\frac{\exp(\beta GD)}{\beta\mu}} \sqrt{3 \log(1/\delta)} = O(1) \frac{GD\beta}{n\sqrt{d}} \sqrt{\exp(\beta GD) \log(1/\delta)} \leq 1$$

and hence prove the privacy guarantee.

As for the empirical risk bound, by Lemma 5.3, we know

$$\mathop{\mathbb{E}}_{x \sim \pi}\left(F_{\mathcal{D}}(x) + \frac{\mu}{2}\|x\|_2^2\right) - \min_{x^* \in \mathcal{K}}\left(F_{\mathcal{D}}(x^*) + \frac{\mu}{2}\|x^*\|_2^2\right) \lesssim \frac{d \log(\beta GD/d)}{\beta},$$

and we know

$$\mathop{\mathbb{E}}_{x \sim \pi} F_{\mathcal{D}}(x) - \min_{x^* \in \mathcal{K}} F_{\mathcal{D}}(x^*) \lesssim \frac{d \log(\beta GD/d)}{\beta} + \mu D^2.$$

Replacing the value of $\beta$ achieves the empirical risk bound.

As for the population risk, we have

$$
\begin{aligned}
&\underset{x\sim\pi}{\mathbb{E}}\,F_{\mathcal{P}}(x) - \min_{y^*\in\mathcal{K}}F_{\mathcal{P}}(y^*) \\
&= \underset{x\sim\pi}{\mathbb{E}}\,[F_{\mathcal{P}}(x) - F_{\mathcal{D}}(x)] + \mathbb{E}[F_{\mathcal{D}}(x) - \min_{x^*\in\mathcal{K}}F_{\mathcal{D}}(x^*)] + \mathbb{E}[\min_{x^*\in\mathcal{D}}F_{\mathcal{D}}(x^*) - \min_{y^*\in\mathcal{K}}F_{\mathcal{P}}(y^*)] \\
&\leq \underset{x\sim\pi}{\mathbb{E}}\,[F_{\mathcal{P}}(x) - F_{\mathcal{D}}(x)] + \mathbb{E}[F_{\mathcal{D}}(x) - \min_{x^*\in\mathcal{K}}F_{\mathcal{D}}(x^*)].
\end{aligned}
$$

We can bound $\mathbb{E}_{x\sim\pi}[F_{\mathcal{P}}(x) - F_{\mathcal{D}}(x)] \leq O(\frac{G^2\exp(\beta GD)}{n\mu}) \leq O(\frac{GD\varepsilon\log(n)}{n^{1-c}d\sqrt{\log(1/\delta)}})$ by Lemma 5.5 for an arbitrarily small constant $c > 0$. Hence the empirical risk is dominated term compared to $\mathbb{E}_{x\sim\pi}[F_{\mathcal{P}}(x) - F_{\mathcal{D}}(x)]$, and we complete the proof. $\qquad\square$

### D.3 Implementation

We rewrite them below: Let $\widehat{F}(x) := F(x) + r(x)$ where $r(x)$ is some regularizer, and $F = \mathbb{E}_{i\in I}\,f_i$ is the expectation of a family of $G$-Lipschitz functions.

---

**Algorithm 3** AlternateSample, [35]

---

1: **Input:** Function $\widehat{F}$, initial point $x_0 \sim \pi_0$, step size $\eta$
2: **for** $t \in [T]$ **do**
3: $\quad y_t \leftarrow x_{t-1} + \sqrt{\eta}\zeta$ where $\zeta \sim \mathcal{N}(0, I_d)$
4: $\quad$ Sample $x_t \leftarrow \exp(-\widehat{F}(x) - \frac{1}{2\eta}\|x - y_t\|_2^2)$
5: **end for**
6: **Output:** $x_T$

---

**Theorem D.1** (Guarantee of Algorithm 3, [15]). *Let $\mathcal{K} \subset \mathbb{R}^d$ be a convex set of diameter $D$, and $\widehat{F} : \mathcal{K} \to \mathbb{R}$, and $\pi \propto \exp(-\widehat{F})$ satisfies LSI with constant $C_{\mathrm{LSI}}$. Then set $\eta \geq 0$, we have*

$$
R_q(\pi_t, \pi) \leq \frac{R_q(\pi_0, \pi)}{(1 + \eta/C_{\mathrm{LSI}})^{2t/q}},
$$

*where $R_q(\pi', \pi)$ is the $q$-th order of Renyi divergence between $\pi'$ and $\pi$.*

To get a sample from $\exp(-\widehat{F}(x) - \frac{1}{2\eta}\|x - y_t\|_2^2)$, we use the rejection sampler from [26], whose guarantee is stated below:

**Lemma D.2** (Rejection Sampler, [26]). *If the step size $\eta \lesssim G^{-2}\log^{-1}(1/\delta_{inner})$ and the inner accuracy $\delta_{inner} \in (0, 1/2)$, there is an algorithm that can return a random point $x$ that has $\delta_{inner}$ total variation distance to the distribution proportional to $\exp(-\widehat{F}(x) - \frac{1}{2\eta}\|x - y\|_2^2)$. Moreover, the algorithm accesses $O(1)$ different $f_i$ function values and $O(1)$ samples from the density proportional to $\exp(-r(x) - \frac{1}{2\eta}\|x - y\|_2^2)$.*

Combining Theorem 5.6, Theorem D.1 and Lemma D.2, we can get the following implementation of the exponential mechanism for non-smooth functions.

**Theorem 5.7** (Implementation, risk bound). *For $\varepsilon, \delta \in (0, 1/2)$, there is an $(\varepsilon, 2\delta)$-DP efficient sampler that can achieve the empirical and population risks $O(GD\frac{d\cdot\log\log(n)\sqrt{\log(1/\delta)}}{\varepsilon\log(nd)})$. Moreover, in expectation, the sampler takes $\tilde{O}\left(n\varepsilon^3\log^3(d)\sqrt{\log(1/\delta)}/(GD)\right)$ function values query and some Gaussian random variables restricted to the convex set $\mathcal{K}$ in total.*

*Proof.* By Theorem 5.6, it suffices to get a good sample from $\pi$ with density proportional to $\exp(-\beta(F_{\mathcal{D}}(x) + \frac{\mu}{2}\|x\|_2^2))$ where $\beta = O(\frac{\varepsilon\log(nd)}{GD\sqrt{\log(1/\delta)}}), \mu = \frac{d}{D^2\beta}$. Set $q = 1$, which gives that $R_q(\cdot, \cdot)$ is the KL-divergence. Suppose we let $x_0$ is drawn from density proportional to $\exp(-\frac{\beta}{2}\mu\|x\|_2^2)$, then the KL divergence between $\pi_0$ and $\pi$ is bounded by $\exp(q\beta GD)$.

Now let $\pi_T^{(i)}$ be the distribution we get over $x_T$ from Algorithm 3 if we use an exact sampler for $i$ iterations, then the sampler of Lemma D.2 for the remaining $T - i$ iterations. The output of Algorithm 3 that we actually get is $\pi_T^{(0)}$. Note that $C_{\text{LSI}} \leq D^2 n$, and $\eta \lesssim \beta^{-2} G^{-2} \log^{-1}(2T/\delta)$. Setting

$$T = O\left(\frac{C_{\text{LSI}}}{\eta} \log(\exp(q\beta GD)/\delta^2)\right) = \tilde{O}\left(\frac{n\varepsilon^3 \log^3(d)\sqrt{\log(1/\delta)}}{GD}\right)$$

we get $\delta_{inner} = \delta/2T$ in Lemma D.2 and that $R_1(\pi_T^{(T)}, \pi) \leq \delta^2/8$. This implies the total variation distance between $\pi_T^{(T)}$ and $\pi$ is at most $\delta/2$ by Pinsker's inequality. Furthermore, by the post-processing inequality, the total variation distance between $\pi_T^{(i)}$ and $\pi_T^{(i+1)}$ is at most $\delta/2T$ for all $i$. Then by triangle inequality the total variation distance between $\pi_T^{(0)}$ and $\pi$ is at most $\delta$. $\qquad\square$

### D.4 Proof of Theorem 5.8

**Theorem 5.8.** *There exists an $\varepsilon$-DP differentially private algorithm that achieves a population risk of $O\left(GD\left(d\log(\varepsilon n/d)/(\varepsilon n) + \sqrt{d\log(\varepsilon n/d)}/(\sqrt{n})\right)\right)$.*

*Proof.* We pick a maximal packing $P$ of $O((D/r)^d)$ points, such that every point in $\mathcal{K}$ is distance at most $r$ from some point in $P$. By $G$-Lipschitzness, the risk of any point in $P$ for the DP-ERM/SCO problems over $\mathcal{K}$ are at most $Gr$ plus the risk of the same point for DP-ERM/SCO over $P$. The exponential mechanism over $P$ gives a DP-ERM risk bound of $O\left(\frac{GD}{\varepsilon n} \log|P|\right)$. Next, note that the empirical loss of each point in $P$ is the average of $n$ random variables in $[0, GD]$ wlog. So, the expected maximum difference between the empirical and population loss of any point in $P$ is $O\left(\frac{GD\sqrt{\log|P|}}{\sqrt{n}}\right)$. Putting it all together we get a DP-SCO expected risk bound of:

$$O\left(Gr + GD\left(\frac{d\log(D/r)}{\varepsilon n} + \frac{\sqrt{d\log(D/r)}}{\sqrt{n}}\right)\right).$$

This is approximately minimized by setting $r = Dd/\varepsilon n$. This gives a bound of:

$$O\left(GD\left(\frac{d\log(\varepsilon n/d)}{\varepsilon n} + \frac{\sqrt{d\log(\varepsilon n/d)}}{\sqrt{n}}\right)\right).$$

$\qquad\square$

## E Conclusion

We present a novel framework that can improve upon the state-of-the-art rates for locating second-order stationary points for both empirical and population risks. We also examine the utilization of the exponential mechanism to attain favorable excess risk bounds for both a polynomial time sampling approach and an exponential time sampling approach. Despite the progress made, several interesting questions remain. There is still a gap between the upper and lower bounds for finding stationary points. As noted in [2], it is quite challenging to beat the current $(\frac{\sqrt{d}}{n})^{2/3}$ empirical upper bound, and overcoming this challenge may require the development of new techniques. A potential avenue for improving the population rate for SOSP could be combining our drift-controlled framework with the tree-based private SpiderBoost algorithm in [2]. Additionally, it is worth exploring if it is possible to achieve better excess risk bounds within polynomial time, and what the optimal risk bound could be.

## F Extended related work

In the convex setting, it is feasible to achieve efficient algorithms with good risk guarantees. In turn, differentially private empirical risk minimization (DP-ERM) [13, 14, 17, 28, 33, 9, 43, 41, 42] and differentially private stochastic optimization [4, 7, 6, 22, 34, 3, 32, 26, 23, 12, 27] have been two of

the most extensively studied problems in the DP literature. Most common approaches are variants of DP-SGD [14] or the exponential mechanism [38].

As for the non-convex optimization, due to the intrinsic challenges in minimizing general non-convex functions, most of the previous works [49, 50, 47, 46, 56, 42, 44, 54, 2, 51, 24] adopted the gradient norm as the accuracy metric rather than risk. Instead of minimizing the gradient norm discussed before, [8] tried to minimize the stationarity gap of the population function privately, which is defined as $\text{Gap}_{F_{\mathcal{P}}}(x) := \max_{y \in \mathcal{K}} \langle \nabla F_{\mathcal{P}}(x), x - y \rangle$, which requires $\mathcal{K}$ to be a bounded domain. There are also some different definitions of the second order stationary point. We refer the readers to [37] for more details.

The risk bound achieved by algorithms with polynomial running time is weak and requires $n \gg d$ to be meaningful. Many previous works consider minimizing risks of non-convex functions under stronger assumptions, such as, Polyak-Lojasiewicz condition [49, 55], Generalized linear model (GLM) [46] and weakly convex functions [8].

In the (non-private) classic stochastic optimization, there is a long line of influential works on finding the first and second-order stationary points for non-convex functions, [1, 29, 21, 53, 18].

**First order stationary points.** Progress towards privately finding a first-order stationary point is measured in ($i$) the norm of the empirical gradient at the solution $x$, i.e., $\|\nabla F_{\mathcal{D}}(x)\|$, and ($ii$) the norm of the population gradient, i.e., $\|\nabla F_{\mathcal{P}}(x)\|$. We summarize compare these first-order guarantees achieved by Algorithm 1 with previous algorithms in Table 2:

| References | Empirical | Population |
|---|---|---|
| [49] | $\frac{d^{1/4}}{\sqrt{n}}$ | N/A |
| [47] | $\frac{d^{1/4}}{\sqrt{n}}$ | $\frac{\sqrt{d}}{\sqrt{n}}$ |
| [50] | $(\frac{\sqrt{d}}{n})^{2/3}$ | N/A |
| [56] | $\frac{d^{1/4}}{\sqrt{n}}$ | $\frac{d^{1/4}}{\sqrt{n}}$ |
| [44] | $\frac{1}{\sqrt{n}} + \left(\frac{\sqrt{d}}{n}\right)^{2/3}$ | N/A |
| [2] | $\left(\frac{\sqrt{d}}{n}\right)^{2/3}$ | $\frac{1}{n^{1/3}} + (\frac{\sqrt{d}}{n})^{1/2}$ |

Table 2: Previous work in finding first-order stationary points. We omit logarithmic terms and dependencies on other parameters such as Lipschitz constant. "N/A" means we do not find an explicit result in the work.

**Second order stationary points.** We say a point $x$ is a Second-Order Stationary Point (SOSP), or a local minimum of a twice differentiable function $g$ if $\|\nabla g(x)\|_2 = 0$ and $\text{smin}(\nabla^2 g(x)) \geq 0$. Exact second-order stationary points can be extremely challenging to find [25]. Instead, it is common to measure the progress in terms of how well the solution approximates an SOSP.

**Definition F.1** (approximate-SOSP, [1]). *We say $x \in \mathbb{R}^d$ is an $\alpha$-second order stationary point ($\alpha$-SOSP) for $\rho$-Hessian Lipschitz function $g$, if*

$$\|\nabla g(x)\|_2 \leq \alpha \bigwedge \text{smin}(\nabla^2 g(x)) \geq -\sqrt{\rho\alpha}.$$

| References | Empirical | Population |
|---|---|---|
| [46] | $\frac{d^{1/4}}{\sqrt{n}}$ | N/A |
| [48] | $(\frac{d}{n})^{4/7}$ | N/A |
| [24] | $(\frac{\sqrt{d}}{n})^{1/2}$ | N/A |
| Ours | $(\frac{\sqrt{d}}{n})^{2/3}$ | $\frac{1}{n^{1/3}} + (\frac{\sqrt{d}}{n})^{3/7}$ |

Table 3: Summary of previous results in finding $\alpha$-SOSP, where $\alpha$ is demonstrated in the Table. Omit the logarithmic terms and the dependencies on other parameters like Lipschitz constant. "N/A" means we do not find an explicit result in the work.

Existing works in finding approximate SOSP privately give guarantees for the empirical function $F_{\mathcal{D}}$. We improve upon the state-of-the-art result and give the first guarantee for the population function $F_{\mathcal{P}}$, which is summarized in Table 3.

