\left(\frac{d/\omega}{\zeta_2^2\kappa + \zeta_1^2}\right)} \cdot \log^3\left(\frac{BMd}{\rho\omega(\zeta_2^2\kappa + \zeta_1^2)}\right).$$

As mentioned before, we can bound the number of occurrences where the drift gets large and hence bound the total time we query the oracle of the first kind.

**Lemma 3.7.** *Under the event that $\|\nabla_t - \nabla F(x_t)\| \leq \gamma/4$ for all $t \in [T]$ and our parameter settings, letting $K = \{t \in [T] : \mathrm{drift}_t \geq \kappa\}$ be the set of iterations where the drift is large, we know $|K| \leq O\left(\frac{B\eta}{\kappa} + T\gamma^2\eta^2/\kappa\right) = O\left(B\eta \log^4(\frac{dMB}{\rho\gamma\omega})/\kappa\right).$*

# 4 Private SOSP

We adopt the framework before and get our main results on finding SOSP privately by constructing private gradient oracles in this section. Finding SOSP for empirical risk function $F_\mathcal{D}$ and for population risk function $F_\mathcal{P}$ are discussed in Subsection 4.1 and Subsection 4.2 respectively.

## 4.1 Convergence to the SOSP of the Empirical Risk

We use Stochastic Spider to improve the convergence to $\alpha$-SOSP of the empirical risk, and aim at getting $\alpha = \tilde{O}(d^{1/3}/n^{2/3})$. We use the full-batch size for simplicity, and use the gradient oracles

$$\mathcal{O}_1(x) := \nabla F_\mathcal{D}(x) + g_1, \quad \text{and} \quad \mathcal{O}_2(x, y) := \nabla F_\mathcal{D}(x) - \nabla F_\mathcal{D}(y) + g_2, \tag{2}$$

where $g_1 \sim \mathcal{N}(0, \sigma_1^2 I_d)$ and $g_2 \sim \mathcal{N}(0, \sigma_2^2\|x - y\|_2^2 I_d)$ are added to ensure privacy by Gaussian mechanism (in Appendix).

Before stating the formal results, note that by Lemma 3.6, the framework can only guarantee the existence of an $\alpha$-SOSP in the outputted set. In order to find the SOSP privately from the set, we adopt the well-known AboveThreshold algorithm, whose pseudo-code can be found in Algorithm 2 in the Appendix. Algorithm 2 is a slight modification of the well-known AboveThreshold algorithm in [19], and we get the following guarantee immediately.

**Lemma 4.1.** *Algorithm 2 is $(\varepsilon, 0)$-DP. Given the point set $\{x_1, \cdots, x_T\}$ and $S$ of size $n$ as the input, (i) if it outputs any point $x_i$, then with probability at least $1 - \omega$, we know*

$$\|\nabla F_S(x_i)\| \leq \alpha + \frac{32\log(2T/\omega)G}{n\varepsilon}, \text{ and } \mathrm{smin}(\nabla^2 F_S(x_i)) \geq -\sqrt{\rho\alpha} - \frac{32\log(2T/\omega)M}{n\varepsilon}$$

*(ii) if there exists a $\alpha$-SOSP point $x \in \{x_i\}_{i\in[T]}$, then with probability at least $1 - \omega$, Algorithm 2 will output one point.*

Choosing the appropriate noise scales for the Gaussian added in Equation (2) and running Algorithm 1 can get a private set of points which contains at least one good SOSP. Then we can run Algorithm 2 to find the good SOSP in the set privately. The formal guarantee is stated below:

**Theorem 4.2** (Empirical). *For $\varepsilon \leq 10, \delta \in (0, 1/2)$, use Equation (2) as gradient oracles with $\kappa = \frac{G^{4/3}B^{1/3}}{M^{5/3}}(\frac{\sqrt{d\log(1/\delta)}}{n\varepsilon})^{2/3}$, $\sigma_1 = \frac{G\sqrt{B\eta\log^2(n/\delta)/\kappa}\log^2(ndMB/\omega)}{n\varepsilon}, \sigma_2 = \frac{M\sqrt{\log^2(n/\delta)BM/\alpha_1^2}\log^5(ndMB/\omega)}{n\varepsilon}$. Running Algorithm 1, outputting the set $\{x_i\}_{i\in[T]}$ if the total time to query $\mathcal{O}_1$ is bounded by $O(B\eta\log^4(\frac{dMB}{\rho\gamma\omega})/\kappa)$, otherwise outputting a set of $T$ arbitrary points is $(\varepsilon/2, \delta)$-DP. With probability at least $1 - \omega$, at least one point in the output set is $\alpha_1$-SOSP of $F_{\mathcal{D}}$ with*

$$\alpha_1 = O\left(\left(\frac{\sqrt{dBGM\log^2(1/\delta)}}{n\varepsilon}\right)^{2/3} \cdot \log^6\left(\frac{nBMd}{\rho\omega}\right)\right).$$

*Moreover, if we run Algorithm 2 with inputs $\{x_i\}_{i\in[T]}, \mathcal{D}, B, M, G, \rho, \alpha_1$, with probability at least $1 - \omega$, we can get an $\alpha_2$-SOSP of $F_{\mathcal{D}}$ with $\alpha_2 = O\left(\alpha_1 + \frac{G\log(n/G\omega)}{n\varepsilon} + \frac{M\log(ndBGM/\rho\omega)}{n\varepsilon\sqrt{\rho}}\sqrt{\alpha_1}\right)$. The whole procedure is $(\varepsilon, \delta)$-DP.*

**Remark 4.3.** *It's worth noting that the cost of gradient computation can be reduced by utilizing smaller batch sizes. Additionally, the application of Rényi Differential Privacy techniques may enhance results by some logarithmic terms.

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

**Theorem A.2** (Basic composition, [19]). *If $\mathcal{A}_1$ is $(\varepsilon_1, \delta_1)$-DP and $\mathcal{A}_2$ is $(\varepsilon_2, \delta_2)$-DP, then their combination is $(\varepsilon_1 + \varepsilon_2, \delta_1 + \delta_2)$-DP.*

**Theorem A.3** (Advanced composition, [30]). *For $\varepsilon \leq 0.9$, an end-to-end guarantee of $(\varepsilon, \delta)$-differential privacy is satisfied if a database is accessed at most $k$ times, where each time with a $(\varepsilon/(2\sqrt{2k\log(2/\delta)}), \delta/(2k))$-differentially private mechanism.*

Due to space limit, some other preliminaries and proofs are left in the Appendix.

**Theorem A.4** (Gaussian Mechanism, [19]). *Given a randomized algorithm $\mathcal{A} : P^* \to \mathbb{R}^d$, let $\Delta_2 f = \max_{neighboring\ \mathcal{D},\mathcal{D}'} \|\mathcal{A}(\mathcal{D}) - \mathcal{A}(\mathcal{D}')\|_2$, then adding noise scaled to $\mathcal{N}(0, \sigma^2)$ with $\sigma \geq \frac{\sqrt{2\log(1.25/\delta)}\Delta_2 f}{\varepsilon}$ is $(\varepsilon, \delta)$-DP.*

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

^{5/3}}\big(\frac{\sqrt{d\log(1/\delta)}}{n\varepsilon}\big)^{2/3}$, $\sigma_1 = \frac{G\sqrt{B\eta\log^2(n/\delta)/\kappa}\log^2(ndMB/\omega)}{n\varepsilon}$, $\sigma_2 = \frac{M\sqrt{\log^2(n/\delta)BM/\alpha_1^2}\log^5(ndMB/\omega)}{n\varepsilon}$. Running Algorithm 1, outputting the set $\{x_i\}_{i\in[T]}$ if the total time to query $\mathcal{O}_1$ is bounded by $O(B\eta\log^4(\frac{dMB}{\rho\gamma\omega})/\kappa)$, otherwise outputting a set of $T$ arbitrary points is $(\varepsilon/2, \delta)$-DP. With probability at least $1 - \omega$, at least one point in the output set is $\alpha_1$-SOSP of $F_\mathcal{D}$ with*

$$\alpha_1 = O\left(\left(\frac{\sqrt{dBGM\log^2(1/\delta)}}{n\varepsilon}\right)^{2/3} \cdot \log^6\left(\frac{nBMd}{\rho\omega}\right)\right).$$

*Moreover, if we run Algorithm 2 with inputs $\{x_i\}_{i \in [T]}, \mathcal{D}, B, M, G, \rho, \alpha_1$, with probability at least $1 - \omega$, we can get an $\alpha_2$-SOSP of $F_\mathcal{D}$ with $\alpha_2 = O\left(\alpha_1 + \frac{G \log(n/G\omega)}{n\varepsilon} + \frac{M \log(ndBGM/\rho\omega)}{n\varepsilon\sqrt{\rho}}\sqrt{\alpha_1}\right)$. The whole procedure is $(\varepsilon, \delta)$-DP.*

*Proof.* The privacy guarantee can be proved by composition theorems (Theorem A.2 and Theorem A.3), Gaussian Mechanism (Theorem A.4) and Lemma 3.7. Specifically, by Assumption 3.1 and our settings of parameters, we know the sensitivity of $\mathcal{O}_1$ and $\mathcal{O}_2$ are bounded by $\frac{G}{n}$ and $\frac{M\|x-y\|}{n}$ respectively, and querying $\mathcal{O}_1$ and $\mathcal{O}_2$ each time are $\left(\frac{\varepsilon}{\sqrt{B\eta \log(n/\delta) \log^2(ndMB/\omega)}}, \delta/n^2\right)$-DP and $\left(\frac{\varepsilon}{\sqrt{\log(n/\delta)BM/\alpha_1^2 \log^5(ndMB/\omega)}}, \delta/n^2\right)$-DP respectively. We can apply the advanced composition to prove the privacy guarantee of the whole algorithm. As the total number of iterations $T$ is determined, and the privacy cost to query $\mathcal{O}_2$ for $T$ times is controlled. It suffices to bound the total time to query $\mathcal{O}_1$, which is guaranteed in the statement. That is if the total time to query $\mathcal{O}_1$ is bounded by $O\left(B\eta \log^4(\frac{dMB}{\rho\gamma\omega})/\kappa\right)$, the privacy guarantee follows from the advanced compostition. If the time exceeds $O\left(B\eta \log^4(\frac{dMB}{\rho\gamma\omega})/\kappa\right)$, then we will output a set of arbitrary points which does not occur additional privacy cost.

As for the utility, we know the $\mathcal{O}_1$ and $\mathcal{O}_2$ constructed in Equation (2) are first kind of $\sigma_1\sqrt{d}$ and second kind of $\sigma_2\sqrt{d}$ norm-subGaussian gradient oracle by Fact 2.3. Hence by Lemma 3.6, the utility $\alpha_1$ satisfies that

$$
\begin{aligned}
\alpha_1 &= O(\sigma_1\sqrt{d} + \sigma_2\sqrt{d\kappa}) \cdot \log^3(BMd/\rho\omega) \\
&= O\left(\frac{L\sqrt{dB\eta \log^2(1/\delta)/\kappa}}{n\varepsilon} + \frac{M \log^3(ndMB/\omega)\sqrt{\log^2(1/\delta)BM}}{n\varepsilon\alpha_1}\sqrt{d\kappa}\right) \cdot \log^5(nBMd/\