# OpenReview forum: "Private (Stochastic) Non-Convex Optimization Revisited: Second-Order Stationary Points and Excess Risks"
_NeurIPS.cc/2023/Conference — NeurIPS 2023 spotlight_

### Official Review · Reviewer_xUe2 · 2023-07-03

**Soundness:** 3 good
**Presentation:** 3 good
**Contribution:** 3 good
**Rating:** 7
**Confidence:** 1

**Summary:**

This paper provides a private algorithm for (Stochastic) Non-Convex Optimization using gradients, differences between two successive gradients, and the exponential mechanism for privacy (added to the gradient). It leverages the SpiderBoost algorithm and its private version to find (privately) stationary points. The main challenge addressed by the paper is to keep the quality of the gradient estimations at all steps.

**Strengths:**

- simple modification of  SpiderBoost to maintain the quality of the gradient estimations.
- interesting properties of the algorithm, i.e., the algorithm is capable of escaping the saddle point  with high probability, and a large drift implies significant decrease in the function value, limiting the number of gradient queries.

**Weaknesses:**

- there is something weird about Algorithm 1. drift_t is set to zero in lines 7 and 10, but then overwritten in line 14.

**Questions:**

See above

**Limitations:**

- traditional hypotheses on these types of results

---

> ### Author Rebuttal · Authors · 2023-08-09
>
> Thanks for your time reviewing and commenting on our paper.
> Thanks for pointing out the typo. It should be $\mathrm{drift}_{t+1}=\mathrm{drift}_t+ \eta^2 ||\nabla_t||^2$ in line 14. Please let us know if you have further questions or concerns.

---

> > ### Comment · Area_Chair_JEP2 · 2023-08-19
> > **Respond to authors**
> >
> > Dear reviewer,
> >
> > please read authors' response to your review and reply to them regarding how it changed (or did not change) your evaluation of this work.
> >
> > Thank you in advance,
> >
> > Your AC

---

### Official Review · Reviewer_mRqH · 2023-07-06

**Soundness:** 4 excellent
**Presentation:** 4 excellent
**Contribution:** 3 good
**Rating:** 7
**Confidence:** 4

**Summary:**

This paper addresses the problem of private stochastic optimization for general non-convex functions. For smoothed functions, the authors propose a combination of the Gaussian mechanism and the variance reduction technique (spider algorithm) to find the SOSP of the loss function. Their results for the empirical risk improve upon previous state-of-the-art bounds by a factor of $n\varepsilon$ and their results for the population risk are new in the field of private optimization literature. For general Lipschitz non-convex problems, the authors establish improved bounds for bounding the excess risk using exponential mechanisms. Notably, they provide a new exponentially-time algorithm that matches the proposed lower bound.


**Strengths:**

The paper is well-written and clear in its claims. The authors propose a novel approach to private stochastic optimization for both smooth and non-convex loss functions by combining existing private mechanisms with the Spider algorithm. They provide new results for both upper and lower bounds on the excess population risk, which are technically solid and contribute to the literature on private stochastic optimization. Overall, the results presented in the paper make a valuable contribution to the field.



**Weaknesses:**

1. The paper makes a significant contribution to improving the bounds for private optimization, but there is still a gap between the existing upper and lower bounds that needs to be explored in future work.
2. While the paper focuses on theoretical studies, the authors could consider conducting experimental studies to validate their theoretical findings

**Questions:**

I have no further questions.

**Limitations:**

Yes

---

> ### Author Rebuttal · Authors · 2023-08-09
>
> Thanks for your kind reviews and comments. In our future work, we will both explore the tightness of our bounds, and evaluate our algorithms empirically.

---

> ### Comment · Area_Chair_JEP2 · 2023-08-19
> **Respond to authors**
>
> Dear reviewer,
>
>   please read authors' response to your review and reply to them regarding how it changed (or did not change) your evaluation of this work.
>
> Thank you in advance,
>
>   Your AC

---

### Official Review · Reviewer_52pN · 2023-07-07

**Soundness:** 3 good
**Presentation:** 2 fair
**Contribution:** 3 good
**Rating:** 5
**Confidence:** 2

**Summary:**

The paper introduces a non-convex optimization framework that integrates two types of gradient oracle and an application of the exponential mechanism to find the global minima of non-convex objectives, requiring minimal assumptions and showcasing remarkable population risk bound performance. The paper also highlights the ability to emulate previous empirical and population risk bounds effectively, removing the necessity for smoothness assumptions.

**Strengths:**

This paper presents robust, theoretical insights into a clearly defined problem, and its logical flow makes it accessible and easy to comprehend.

It commendably surveys and incorporates existing related literature.

**Weaknesses:**

This paper appears to conclude abruptly, with the discourse ending unexpectedly following a theorem. The absence of a conclusion or summary weakens the paper's overall structure.

There is an absence of numerical experiments.

Additionally, the authors seem to have relied solely on Advanced Composition for privacy accounting. There are, however, various alternative privacy accounting methods available that could potentially improve the overall performance. For instance, the 'Better Privacy Accounting' method [1], utilization of GDP [2], or the application of tighter composition theorems [3] might be considered for future research and improvements.

[1]: Altschuler, J., & Talwar, K. (2022). Privacy of noisy stochastic gradient descent: More iterations without more privacy loss. Advances in Neural Information Processing Systems, 35, 3788-3800.

[2] Liu, Y., Sun, K., Jiang, B., & Kong, L. (2022). Identification, amplification and measurement: A bridge to gaussian differential privacy. Advances in Neural Information Processing Systems, 35, 11410-11422.

[3] Kairouz, P., Oh, S., & Viswanath, P. (2015, June). The composition theorem for differential privacy. In International conference on machine learning (pp. 1376-1385). PMLR.

**Questions:**

How does the performance of the Stochastic Spider, as presented in this paper, compare to that of the BoostSpider in practical applications? While the theoretical results indicate a rate improvement, is this enhancement accompanied by an increase in constants and an impact on finite sample performances?

**Limitations:**

See weaknesses.

---

> ### Author Rebuttal · Authors · 2023-08-09
>
> Thank you for your comments. We will add a detailed conclusion section in the paper. In more detail, our conclusion will review the main contributions of the paper and then propose some future directions which we feel are important/interesting, including:
> - Can we close the gap between our upper and lower bounds for SOSP?
> - We used a “gradient difference oracle” O_2, which has lower sensitivity, and thus we can tolerate a higher noise multiplier on this oracle. Are there other settings where such a technique can improve the privacy-utility tradeoff?
> - Can we obtain our SOSP bounds using a batch gradient oracle instead of a full gradient oracle?
> - For population risk, do any assumptions “between” convexity and LSI admit better excess loss bounds in polynomial time?
>
> Comment about the numerical experiment: The objective of our paper was to improve the current SOTA analytical bounds (see [1] and [2] in response to Reviewer nTrG, which are also purely theoretical in nature). We acknowledge having a thorough empirical study is important. We leave it for future exploration.
>
> Our results seamlessly extend to measures of privacy like GaussianDP/ RenyiDP/ zCDP.  We will update the presentation to state the main results in terms of Gaussian DP and using Gaussian DP composition (as opposed to ($\epsilon,\delta$)-DP). One reason for choosing the ($\epsilon,\delta$)-DP variant was to be consistent with the prior literature. We want to emphasize that changing the notion of privacy does not make any meaningful changes to our results, as (i) our results are based solely on the Gaussian mechanism and AboveThreshold algorithm (which are known to satisfy all the above notions of DP seamlessly.) (ii) we give polynomial improvements on the dependence on n and d, so improving log factors via tighter composition does not qualitatively change the results.

---

> > ### Comment · Reviewer_52pN · 2023-08-18
> >
> > Thank you for your response.  I believe the authors will add a detailed conclusion section and update the presentation to state the main results in terms of Gaussian DP and using Gaussian DP composition. I have adjusted my score to 5 accordingly.

---

### Official Review · Reviewer_nTrG · 2023-07-09

**Soundness:** 3 good
**Presentation:** 3 good
**Contribution:** 4 excellent
**Rating:** 8
**Confidence:** 3

**Summary:**

This paper studies differentially private second-order optimization problems, where the loss function is assumed to be second-order smooth. The objective is to identify second-order stationary points (SOSP) while ensuring privacy guarantees. The authors present results for both empirical risk and population risk, with notable advancements over existing state-of-the-art outcomes. Specifically, the empirical risk minimization (ERM) bound improves from the state-of-art rate, while the population risk result stands as the first of its kind in the literature on differentially private SOSP optimization. Furthermore, the paper contributes a lower bound analysis for private SOSP optimization, further strengthening its significance in this domain.

**Strengths:**

The strength of this paper lies in its highly comprehensive study of private second-order optimization. The authors study both empirical risk minimization and population risk, and the obtained bounds improves from the state-of-the-art results in both case. The inclusion of a lower bound also closes the gap of this problem.

**Weaknesses:**

The authors can enhance the writing by providing additional discussion regarding the algorithm employed, the obtained results, and the insights derived from their findings. Furthermore, providing more detailed explanations for the technical results cited from previous works, such as Lemma 3.4 and Alg.2, would help clarify any confusion and prevent ambiguity.

**Questions:**

I have a few questions regarding the general SOSP framework (Algorithm 1):
- What role does $\text{frozen}_t$ play?
- line 8: why is an additional noise $g_t$ added to the private oracle $\mathcal{O}_1(x_t)$?
- Since Alg. 1 only serves as a non-private SOSP optimization framework, is it possible to choose other variance-reduction algorithms (e.g., [1])? If not, what is the key characteristic that distinguishes SpiderBoost from other optimizers in the private setting?

Also, Section 5 focuses on private excess risk minimization, which seems to deviate from the previous discussion about SOSP. I'm wondering how Sec. 5 relates to the previous part of the paper?

Reference:
1. Ashok Cutkosky, Francesco Orabona. Momentum-Based Variance Reduction in Non-Convex SGD.

---

> ### Author Rebuttal · Authors · 2023-08-09
>
> Thanks for your positive reviews and questions. Following are the responses to the specific questions:
>
> Q1. We use Lemma 3.4 for escaping the saddle point directly, which requires us to add “extra” Gaussian noise whenever the gradient norm of the current point is small (meaning it is possibly a saddle point). Notice that Section 3 discusses the general optimization framework and has nothing to do with DP, and hence the oracle does not necessarily need to add Gaussian noise and can even be exact.
> frozen_t is used to control the “extra” Gaussian noise we add to escape saddle points, as adding too much extra Gaussian noise may worsen the algorithm's performance. For example, when the initial point is already a good SOSP, then ${\rm frozen}_t$ can ensure that further extra noise can be added at most every $\Gamma$ steps.
>
> Q2: As we mentioned in Q1, the Gaussian $g_t$ is used for helping the point escape the saddle point. (A similar idea was used in [2,3]).
>
> Q3: This is also asked by other reviewers, and we will add a discussion on this. We chose SpiderBoost mainly because [1] used it for finding first-order stationary points privately.  We examined whether this algorithm can be used to find second-order stationary points (SOSP), and eventually solved the challenges towards achieving SOSP.
>
> It is true that current Section 5 can be considered as an interlude. (We will make it clear in the paper.) Prior works ([1] and [2]) have considered zeroth order (excess risk), FOSP, and SOSP. Being consistent with the literature, we improve on excess risk bounds too.
>
> [1] Arora, R., Bassily, R., González, T., Guzmán, C. A., Menart, M., & Ullah, E. (2023, July). Faster rates of convergence to stationary points in differentially private optimization. In International Conference on Machine Learning (pp. 1060-1092). PMLR.
>
> [2] Wang, D., Chen, C., & Xu, J. (2019, May). Differentially private empirical risk minimization with non-convex loss functions. In International Conference on Machine Learning (pp. 6526-6535). PMLR.
>
> [3] Jin, C., Ge, R., Netrapalli, P., Kakade, S. M., & Jordan, M. I. (2017, July). How to escape saddle points efficiently. In International conference on machine learning (pp. 1724-1732). PMLR.

---

> > ### Comment · Reviewer_nTrG · 2023-08-15
> >
> > I would like to thank the authors for detailed response, and I remain my rating.

---

### Official Review · Reviewer_kK73 · 2023-07-25

**Soundness:** 3 good
**Presentation:** 2 fair
**Contribution:** 3 good
**Rating:** 7
**Confidence:** 3

**Summary:**

The paper studies the problem of non-convex optimization under differential privacy constraints. The goal of the paper is to find first and second order stationary points of the function. The proposed algorithm which is inspired by SpiderBoost [1] is a sampling based method that generates a private estimator that minimizes both excess empirical and population risks.

The authors studied the cases: when only polynomial run time is allowed, and when the exponential run time is permitted. In the first case, their proposed algorithm outperforms the previous best and tightens the gap between the provided upper bound and the already existing lower bound. For the case when exponential run time is allowed, the paper provides a nearly matching upper and lower bound.

[1] Zhe Wang, Kaiyi Ji, Yi Zhou, Yingbin Liang, and Vahid Tarokh. Spiderboost and momentum: Faster variance reduction algorithms. Advances in Neural Information Processing Systems, 32, 2019.



**Strengths:**

The paper is a collection of elegant results. From a technical point of view, the paper is solid. It enhances advanced techniques to fulfill its objective. While the paper outperforms previous best algorithm [2], with polynomial run time it relaxes the smoothness assumption that has been assumed in [2].

[2] Di Wang, Changyou Chen, and Jinhui Xu. Differentially private empirical risk minimization with non-convex loss functions. In International Conference on Machine Learning, pages 6526–6535. PMLR, 2019.



**Weaknesses:**

For a reader who is not fluent with the literature, the paper is not easy to follow. Several insights left to be unexplained. The paper is quite dense without enough provided intuition on the proposed algorithm. The authors did a good job in mentioning the similarities with precious work, but they did not discuss their technical novelties. Some expressions and notation are left undefined (ex. W_2 at line 308).

As a side note, I think P at line 18 should be P*.


**Questions:**

1. What is the main technical novelty of the work?

2. Is the proposed algorithm a privatized version of SpiderBoost? if yes, except the added gaussian noise what are the additional elements that are added? And can one similarly construct a privatized version of other algorithms like Spider or SARAH?

3. Can you please provide an intuition that why your technique could relax the smoothness assumption?

4. Except the relaxed smoothness assumption, can you please provide an intuition what would be the difference in the perfromance if one used Spider or SARAH instead of SpiderBoost?



**Limitations:**

The limitation of the work is the optimality gap between the upper and lower bound, where the authors mentioned this and referred to it as an open problem.

---

> ### Author Rebuttal · Authors · 2023-08-09
>
> Thanks for your valuable reviews.
>
> We use $P^*$ as the input dataset can be of arbitrary size. For example, we consider a set D of size n in the paper, meaning $D \in P^n$.
>
> Q1: We discuss our techniques and the challenge of previous techniques in Subsection 1.2 (our techniques) and related work (page 2), and more details can be found there. In particular, our main technical ideas are: 1.To use the notion of drift to decide when to add the noise in SpiderBoost to get good gradient estimations all the time, which enables it to escape saddle points to reach SOSP, and 2. Design privacy analysis of the exponential mechanism and sampling method based on log-Sobolev-inequality (as opposed to smoothness) to obtain the excess population risk guarantees.
>
> Q2: Yes, our algorithm can be treated as a privatized version of SpiderBoost.  Besides adding noise, the main added element is the method for choosing when to recompute the gradient from scratch; SpiderBoost does this after a fixed number of rounds, whereas we dynamically choose to do this based on the quantities frozen and drift. We did not try to construct private versions of other algorithms like Spider or SARAH, but we believe our methods and intuitions can also be helpful for those algorithms. However, we do not expect the theoretical bounds there to be better, as all these algorithms are morally similar.
>
> Q3: Sorry for the confusion. For context, we only relax the smoothness assumption for the excess risk, but not for finding the SOSP. The intuition for relaxing the assumption for SOSP is that we are directly providing our bound via LSI (log-Sobolev inequality), rather than via a run of LMC, which explicitly requires smoothness for convergence.
>
> Q4: SpiderBoost was used by [1] to obtain DP-FOSP, and it was natural to seek whether it is able to find DP-SOSP. SpiderBoost allows a larger learning rate and hence may have better practical performance. Empirical evaluation to compare these different algorithms is left for future work.
>
> [1] Arora, R., Bassily, R., González, T., Guzmán, C. A., Menart, M., & Ullah, E. (2023, July). Faster rates of convergence to stationary points in differentially private optimization. In International Conference on Machine Learning (pp. 1060-1092). PMLR.

---

> > ### Comment · Reviewer_kK73 · 2023-08-16
> >
> > I would like to thank the authors for their response. I maintain the score that I gave.

---

### Author Rebuttal · Authors · 2023-08-09

**General Reply:** We thank all the reviewers for their reviews and comments. We will diligently address all the presentation issues raised by the reviewers and enhance the writing quality in future versions, like providing more explanations on the intuitions. In the following, we respond to the specific questions raised by the reviewers.

---

### Decision · Program_Chairs · 2023-09-21

**Decision:**

Accept (spotlight)

**Comment:**

Private stochastic optimization for general non-convex functions is a hard task to which the authors provide solution with analysis of both upper and lower bounds.